# Automated optimisation of solubility and conformational stability of antibodies and proteins

Angelo Rosace[1,2,8,9], Anja Bennett [1,3,4,9], Marc Oeller [1], Mie M. Mortensen[5,6], Laila Sakhnini[1,7], Nikolai Lorenzen[7], Christian Poulsen[3] & Pietro Sormanni [1] ✉

Biologics, such as antibodies and enzymes, are crucial in research, biotechnology, diagnostics, and therapeutics. Often, biologics with suitable functionality are discovered, but their development is impeded by developability issues. Stability and solubility are key biophysical traits underpinning developability potential, as they determine aggregation, correlate with production yield and poly-specificity, and are essential to access parenteral and oral delivery. While advances for the optimisation of individual traits have been made, the co-optimization of multiple traits remains highly problematic and time-consuming, as mutations that improve one property often negatively impact others. In this work, we introduce a fully automated computational strategy for the simultaneous optimisation of conformational stability and solubility, which we experimentally validate on six antibodies, including two approved therapeutics. Our results on 42 designs demonstrate that the computational procedure is highly effective at improving developability potential, while not affecting antigen-binding. We make the method available as a web-server at www-cohsoftware.ch.cam.ac.uk.

Over the past few decades, protein-based biologics have risen to become a key class of therapeutic molecules[1,2], as they offer a range of favourable characteristics, including high specificity, low toxicity and immunogenicity, and the possibility of replacing or supplementing endogenous proteins and hormones[3]. Furthermore, antibodies are also key reagents in biomedical research and diagnostics, and recombinant enzymes find crucial applications in industrial biotechnology.

However, proteins and antibodies destined to research, diagnostic, biotechnology, and especially therapeutic applications are required to endure a wide range of stresses related to manufacturing, development, shipping, storage, and administration. As these stresses are not present in vivo, natural proteins and antibodies have not evolved to withstand them[4].

Therapeutic applications in particular, have stringent requirements, which include high biological activity at relatively low administration dosage and frequency, high concentration formulations, and long product shelf-life[5]. Compared to intravenous delivery, subcutaneous delivery of biotherapeutics has many advantages for patients and reduces healthcare costs. However, only about one third

[1]Centre for Misfolding Diseases, Yusuf Hamied Department of Chemistry, University of Cambridge, Lensfield road, CB2 1EW Cambridge, UK. [2]Master in Bioinformatics for Health Sciences, Universitat Pompeu Fabra, Barcelona, Catalonia, Spain. [3]Department of Mammalian Expression, Global Research Technologies, Novo Nordisk A/S, Novo Nordisk Park 1, 2760 Måløv, Denmark. [4]BRIC, Faculty of Health and Medical Sciences, University of Copenhagen, Ole Maaløes Vej 5, 2200 Copenhagen, Denmark. [5]Department of Purification Technologies, Global Research Technologies, Novo Nordisk A/S, Novo Nordisk Park 1, 2760 Måløv, Denmark. [6]Faculty of Engineering and Science, Department of Biotechnology, Chemistry and Environmental Engineering, University of Aalborg, Fredrik Bajers Vej 7H, 9220 Aalborg, Denmark. [7]Department of Biophysics and Injectable Formulation 2, Global Research Technologies, Novo Nordisk A/S, Måløv 2760, Denmark. [8]Present address: Institute for Research in Biomedicine (IRB), The Barcelona Institute of Science and Technology, Barcelona, Catalonia, Spain. [9]These authors contributed equally: Angelo Rosace, Anja Bennett. ✉e-mail: ps589@cam.ac.uk

of currently approved antibodies is administered in this way[6]. A major challenge with this delivery route is the need to formulate antibodies at high concentrations (up to 200 mg/mL for some approved antibodies) to enable the delivery of the required dose in small injection volumes (-0.5–2.0 mL)[6]. Consequently, insufficient product solubility and the occurrence of aggregation are bottlenecks preventing subcutaneous delivery.

Emerging directions in biotherapeutic development, where progress may provide broad benefits, include oral delivery and inhalation delivery[7]. These administration routes would be highly convenient for patients, and may facilitate the targeting of specific organs such as the gastrointestinal tract, the lungs, and potentially the brain. However, insufficient stability of antibodies is a major obstacle to oral and inhalation delivery, including denaturation by shear stress in nebulization or low pH in the stomach, and degradation by proteases in the lung or digestive and microbial enzymes[7]. Therefore, high conformational stability and solubility will be key requirements for next-generation antibody drugs[7,8].

Solubility and conformational stability are among the most important properties underpinning the developability potential of biologics, which is defined as the likelihood of a drug candidate with suitable functionality to be developed into a manufacturable, stable, safe, and effective drug that can be formulated to high concentrations while retaining a long shelf-life[9–11]. Poor solubility is a major bottleneck for manufacturing[10,12] and has been linked to increased binding polyreactivity[13,14], which is emerging as a key determinant of attrition in clinical development[15]. Similarly, high conformational stability is essential to ensure efficacy and safety during manufacturing, formulation, storage, shipping, and administration[10,16], and various studies have reported strong correlations of stability and solubility with production yield[17–20].

Solubility and conformational stability also determine colloidal stability, that is the long-term integrity of a formulation, through their link with aggregation. Self-association can be triggered via two main pathways. In one, aggregation hotspots on molecular surfaces drive the initial intermolecular assembly, forming aggregates that then may act as seeds to drive further aggregation and may also increase solution viscosity[21,22]. In the other, the presence of partially or fully unfolded states lead to the transient exposure of hydrophobic patches that can elicit the formation of misfolded aggregates[23]. Therefore, maximising solubility and conformational stability can be expected to translate into reduced aggregation propensity, which is perhaps the most common of the degradation reactions that a protein can experience during its biotechnological or therapeutical development[24]. Unfolded and aggregated biologics not only loose activity, but have been reported to be associated with increased risk of inducing immunogenicity upon injection[25–27]. Therefore, regulatory agencies require formulations with minimal amount of aggregates at the end of the formulation shelf-life to grant market approval[28,29].

Taken together, the stresses that biologics must endure, and especially the requirements they need to meet, imply that their biophysical properties must often be optimised far beyond the typical values of natural proteins and antibodies[5]. While methods of in vitro directed evolution are routinely employed to optimise binding affinity, the simultaneous optimisation of multiple biophysical traits remains problematic[30]. As the fundamental forces that drive protein folding, aggregation, and binding are the same, such traits are often conflicting, in the sense that mutations that improve one of them tend to worsen the others[5,31–33].

This process of multi-traits optimization can be compared to solving a Rubik's cube, where each face represents one biophysical property. Changing one face will affect other faces, often detrimentally, and solving a single face is much simpler than completing the puzzle. In protein engineering, it remains highly challenging to select mutations that selectively improve properties of interest while leaving the others unaffected, and there is an unmet need to develop technologies that enable the simultaneous optimisation of different traits.

Computational approaches offer a promising avenue to generate such technologies, as they allow for highly controlled parallel screenings of multiple biophysical properties. Although the problem may in principle be addressable experimentally through massive spending on molecule screening campaigns, this strategy would not be advisable from an environmental standpoint, and companies and laboratories worldwide are increasingly scrutinised on this aspect. Conversely, computer calculations are rapid, inexpensive, and have no material requirements, while their ability to pinpoint specific mutations reduces the environmental impact of downstream experiments by massively lowering the number of candidates for screening. Taken together, these advantages make the implementation of computational methods in antibody and protein development pipelines particularly attractive.

In this work, we contribute to addressing this challenge by introducing a fully automated computational strategy for the simultaneous optimisation of solubility and conformational stability, which have been reported to be conflicting in some cases[5,34–36]. The pipeline works by removing surface-exposed aggregation hotspots leading to poor solubility, and by proposing mutations expected to increase conformational stability and solubility, thus decreasing the population of partially or fully unfolded states in solution. The approach leverages phylogenetic information to reduce false positive predictions and to prevent the modification of functionally relevant sites. Then, the pipeline relies on the CamSol method[37] to carry out predictions of solubility changes upon mutation, and on the energy function Fold-X[38] for the predictions of the associated stability changes. Both algorithms have separately been validated on experimental data for a wide range of proteins and antibodies (CamSol Refs. 4,14,23,37,39–44 and FoldX Refs. 19,34,45–48). Our method also includes an ad-hoc recipe to obtain and exploit suitable phylogenetic information for immunoglobulin variable domains, as these are a key class of biologics that, because of their modular nature, cannot be handled with standard tools for searching homologs.

We validate our approach experimentally by using it to design mutational variants of six different antibodies: three nanobodies and three single-chain variable fragments (scFv), two of which are approved therapeutics.

## Results

The goal of the computational pipeline is to design protein or antibody variants by predicting combinations of mutations that increase both conformational stability and solubility, or one of the two without affecting the other. The method relies on the knowledge of the native structure of the target protein or on the availability of a good structural model, and on a multiple sequence alignment (MSA) of homologous sequences. A position specific scoring matrix (PSSM) is then extracted from the MSA, to provide information on the frequency of the amino acids observed among homologous proteins at each position of the input structure.

The algorithm is built to handle input proteins consisting of multiple polypeptide chains, as well as bound structures where the binding partner can be excluded from the design. Users are also allowed to provide additional input parameters as explained below in the *Algorithm Pipeline* section.

### Phylogenetic information reduces false positive predictions of stability change

The phylogenetic information is used in the pipeline to enable the identification of candidate mutations based on their observed frequencies in natural variants of the protein or antibody under scrutiny. Therefore, these mutations are more likely to be well-tolerated, and thus less likely to disrupt stability[49]. We hypothesized that restricting

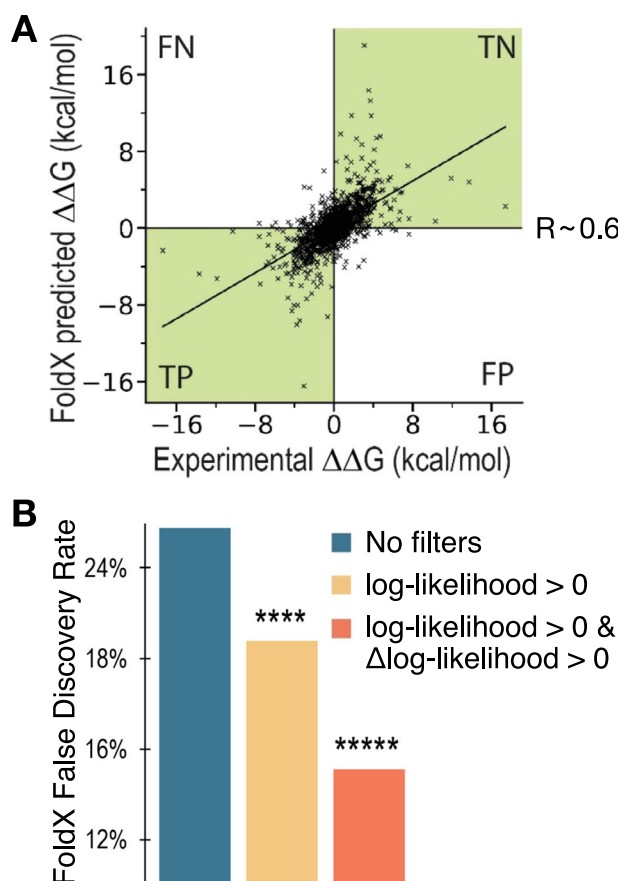

**Fig. 1 | Phylogenetic information reduces false discovery rate. A** Comparison of the experimental ΔΔG against the FoldX-predicted ΔΔG values. Labels are applied to the four quadrants of the graph. (FN: False Negatives, TN: True Negatives, TP: True Positives, FP: False Positives). The Pearson correlation coefficient (R) for the dataset is also shown. The green quadrants identify mutations whose overall effect (stabilizing or destabilizing) is correctly predicted. **B** Bar plot of the FDR of the FoldX prediction obtained by applying different filtering based on phylogenetic information (see legend). The PSSMs employed were obtained with HHblits[57] using a coverage parameter of 60 and identity of 95. Statistical significance was calculated explicitly with random resampling (see Supplementary Methods, ****$p < 0.0001$ and *****$p < 0.00001$).

the mutational space to those mutations that are enriched in natural variants will greatly decrease the number of False Positive (FP) predictions, which can be quantified through the False Discovery Rate (FDR, see Supplementary Methods).

For stability predictions, we define False Positives as those mutations predicted to be stabilizing while being destabilizing in practice, and False Negatives (FN) as the opposite (see Fig. 1A). In the context of computational design, and in particular of algorithms aimed at yielding mutational variants to improve biophysical properties, FNs can be regarded as missed opportunities. These mutations will not be suggested by the algorithm, even if they would be beneficial. However, provided that at least some beneficial mutations are identified, FNs are not the main concern. FP mutations on the other hand would be suggested by the algorithm, leading to a potential waste of time and resources in the experimental production and characterization of designs containing such prediction mistakes. It is therefore of paramount importance that a method for the automated design of mutational variants has the lowest possible FDR[50].

We decided to assess the performance using a recently published database of experimentally characterized mutations[51], which is a curated subset of the ProTherm database where inaccuracies and

biases have been removed[51–53]. This database contains thermodynamic information on the stability changes caused by 755 point mutations within 81 different proteins, and it was developed for the specific purpose of benchmarking computational methods of predicting stability changes[51].

The results of our analysis show that the FDR of the FoldX energy function can be decreased in a statistically significant way by incorporating filters based on phylogenetic information (Fig. 1B). More specifically, we have screened different parameters for the search of homologous sequences and different implementations of the PSSM (see Supplementary Methods and Fig. S1). Our results show that the FDR of the stability predictions can be decreased from ~26% to ~21% ($p < 0.0001$) by restricting the search space to mutations with positive log-likelihood, that is mutations that are observed at that position more often than their background probability (i.e., more often than expected by random chance). The FDR can then be further improved to ~15% ($p < 0.00001$) by only considering mutations that both have a positive log-likelihood and increase the frequency over that of the WT residue (i.e., positive Δlog-likelihood, Fig. 1B). These results are in line with strategies of consensus designs[54,55], as well as with previously reported findings[18,19]. All p-values were calculated explicitly through random resampling (see Supplementary Methods), which demonstrates that it is the specific choice of phylogenetic filtering that is behind this performance improvement, and not a generic restriction of the database size. We also verified that the observed performance improvement is not a consequence of simply removing many of those mutations with predicted ΔΔG close to 0, which would be within the expected error of FoldX (see Supplementary Methods and Fig. S3).

We further verified that similar results can be obtained when using a PSSM containing only raw frequency counts, as opposed to log-likelihood scores (Fig. S1). Such PSSM of raw frequencies is often referred to as Position Probability Matrix (PPM) or Position Weight Matrix (PWM). While log-likelihood scores are generally preferable, as they correct the observed frequencies for the expected background probability of observing each amino acid by random chance, their calculations are often unreliable for alignments containing small numbers of sequences (e.g., <50). Therefore, in cases where the input protein only has a few homologs, the employment of a PWM provides more reliable information, and our algorithm automatically switches to it when less than 50 sequences are contained in the input MSA.

We note that a similar analysis was not possible for the solubility prediction, as a dataset of experimentally measured solubility changes upon mutation, which is large enough to draw any statistically significant conclusions on changes in FDR, is not currently available. Moreover, while all globular proteins must retain good conformational stability to function, one may expect to find an evolutionary pressure towards maintaining high solubility only for those proteins that are expressed to high concentrations, suggesting that the phylogenetic filtering we implemented may not be as beneficial for solubility predictions[23,56]. However, the typical performance of the CamSol method in ranking protein and antibody mutational variants is high (Pearson $R \geq 0.9$)[4,14,37,39,40,43], thus indicating that the FDR of CamSol predictions is already low.

In summary, implementing phylogenetic filtering can reduce the FDR of stability predictions by 11%, at the expense of being left with a smaller—but typically large enough—mutational space to sample. A reduced mutational space means that sometimes potentially beneficial mutations will be left out, but also that the overall pipeline will run much faster, as it only needs to sample a sub-region of the mutational space that is evolutionarily grounded.

## Algorithm pipeline

The method is implemented as a webserver (*www-cohsoftware.ch.cam.ac.uk/index.php/camsolcombination*). Therefore, the user interacts with it through an input form.

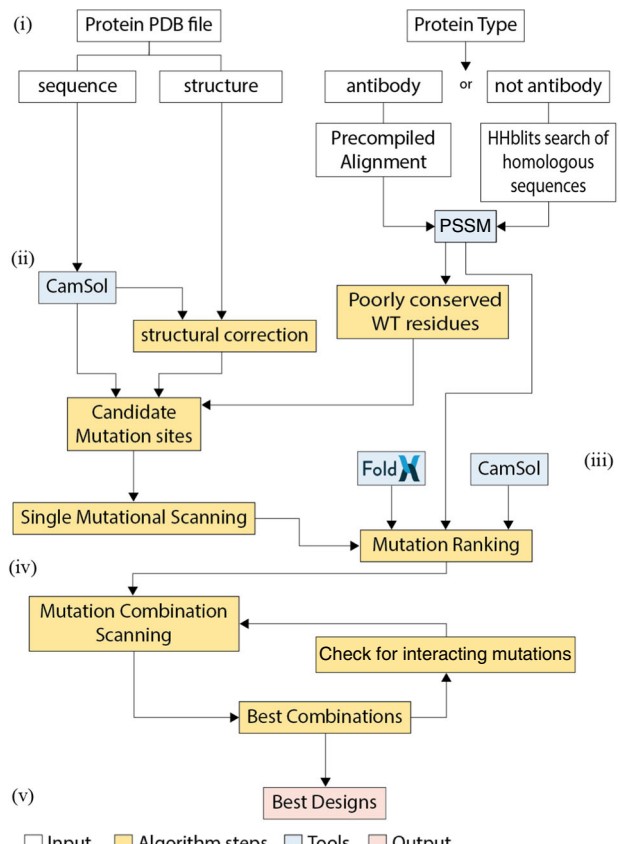

**Fig. 2 | Schematic representation of the algorithm pipeline.** Roman numbers refer to the subsections of the algorithm pipeline section in the main text. Coloured in white are the input processing steps, in blue the tools included in the algorithm, in yellow the core algorithm steps, and in red the output.

**Input processing.** The required input consists of a protein structure in pdb format, its type (antibody Fv region or other protein), and the alignment file(s) to use in the pipeline. If the input protein is not an antibody or nanobody, the user must provide .hhm or.a3m files obtained with HHblits for each chain in the input protein that is not manually excluded from the design. Such files may be obtained by running a HHblits search with a local installation[57], or more straightforwardly from the MPI Bioinformatics Toolkit webserver[58] by following the instructions linked on the CamSol Combination webserver homepage.

If the input protein is an antibody or nanobody, the user can choose between a set of pre-compiled alignments of species-specific antibodies (human or mouse), therapeutic antibodies (post phase-I clinical trial)[59,60], or single-domain antibodies obtained from multiple camelid species[61] (see Supplementary Methods). The need for special alignments for antibodies is due to their modularity, and high degree of conservation in the framework regions coupled with high variability in the binding loops. These features make standard alignment tools unreliable for antibodies, as an excessive number of gaps is typically introduced in loop regions. Furthermore, especially when antibodies are destined to therapeutic applications, it is important to be able to select candidate substitutions that are observed in the relevant species (e.g., from sequences of human origins) to reduce the chances of introducing immunogenic motifs. Similarly, for single-domain antibodies, candidate substitutions must be selected based on alignments of VH sequences known to be able to autonomously fold and remain soluble in the absence of a VL partner.

The user can also provide optional parameters to the design calculations. These include residue positions or whole chains to exclude

from the design, the maximum number of mutations to try simultaneously, and residues to exclude from the list of potential substitution targets. For example, the default procedure does not introduce certain chemical degradation hotspots such as cysteines and methionines, as these are known oxidation sites, and solvent-exposed cysteines may trigger covalent dimerization. Users may also consider excluding asparagines, as these are often prone to deamidation or glycosylation.

The algorithm automatically identifies groups of identical chains that may be present in the input structure. In this way, different mutations are never suggested on chains that are identical in sequence, as in practice such chains would be encoded by the same gene. Identical chains are identified from the *seqres* field in the header of the input PDB file, and from the *atom* sequence only if the *seqres* field is not present. In the PDB format, the *seqres* sequence corresponds to the sequence that was used in the structural determination experiment, while the *atom* sequence is the one for which 3D coordinates have been obtained. The two sequences may differ for example when there are regions of missing electron density. An option is also provided to manually input sequence identity specifications, as groups of chain IDs of identical polypeptide chains, to offer maximum flexibility to accommodate 'non-canonical' PDB structures.

When started, the algorithm first calculates a log-likelihood Position Specific Scoring Matrix (PSSM) from the MSA (Fig. 2). If the number of aligned sequences is below 50, a simpler raw frequency PWM is used instead. The PSSM provides information on which residues are most conserved at each position in the protein sequence, and which mutations are observed in natural variants at each site.

As an example, Supplementary file 1 is the final report of a run on bacillus licheniformis alpha-amylase (PDB ID 1bli). The PSSM plotted therein, calculated from an alignment of homologs obtained from the HHblits webserver by simply uploading the sequence[58], reveals that the active site of the enzyme (Asp 231, Glu 261 and Asp 328) is highly conserved, and so are known ion-binding sites on its surface. Given the high conservation in the PSSM, no mutation would be tried at these positions in the automated pipeline. This example shows that incorporating phylogenetic information can preserve functionally relevant residues in automated computational design pipelines, without the need to manually exclude them from the calculation, which, depending on the case, may require high domain expertise.

**Selection of candidate mutation sites.** After processing the input, the algorithm automatically identifies candidate mutation sites. Candidate sites for mutation are identified based on their contribution to solubility, as predicted with the CamSol method[37], and their level of conservation, as obtained from the PSSM (Fig. 2).

First, CamSol is used to calculate intrinsic solubility profiles for each chain of the input structure. These profiles associate each residue with a number that reflects its impact on the overall solubility[23,37]. This calculation relies solely on the knowledge of the amino acid sequence, which is extracted from the *seqres* field of the input pdb file (if present, otherwise from the *atom* sequence) to account for the contributions to the solubility of possible regions with unassigned 3D coordinates. CamSol predictions rely on a combination of physicochemical properties of amino acids, including charge, hydrophobicity, and propensity to form secondary structure elements[37]. These properties are first considered at the individual residue level, then averaged locally across sequence regions to account for the influence of neighbouring residues, and finally considered globally to yield a solubility score[23,39]. This score, which is one number for the whole sequence or multiple sequences in the case of a complex[39], will later be used to rank different mutational variants. Intrinsic profiles are used to identify candidate mutation sites (i.e., residues in the sequence) that contribute strongly to poor solubility of the protein sequence. At this step, only those sites that are solvent-exposed on the protein surface are flagged as possible candidate mutation sites, as buried poorly soluble residues are

expected within the hydrophobic core of globular proteins. More specifically, residues are considered solvent-exposed if they have a solvent exposure of at least 10%, calculated by dividing the observed solvent-accessible surface area of each residue by that of the amino acid type under scrutiny in the context of a fully extended Gly-Xxx-Gly three-peptide[37]. Typically, a relatively low solvent exposure results in structurally corrected solubility scores close to zero, because of the effect of the exposure weight in the CamSol structurally corrected calculations (see Ref. [37]). However, thermal fluctuations may transiently expose such poorly soluble motifs, thus leading to aggregation. It is therefore important to identify poorly soluble sites that are close to the surface and may therefore become exposed through thermal fluctuations. We refer to this group of candidate mutation sites as *identified from the solubility of the sequence*.

Second, the algorithm focuses on predicting potential aggregation hotspots on the protein native state, which are defined as groups of residues on the surface that create patches of poor solubility. The prediction is based on the CamSol structurally corrected calculation as described in detail in Ref. [37]. In this calculation, the intrinsic profiles are modified to account for the proximity of amino acids in the 3D structure and for their solvent exposure. The corrected profiles are used to identify those mutation sites that contribute most to the overall aggregation propensity of possible surface hotspots. We refer to this group of candidate mutation sites as *identified from the solubility of the structure*.

Third, also those sites that are relatively well conserved (with conservation index greater than 0.25, see Supplementary Methods), and where the frequency of the WT residue is low (log-likelihood ≤ 0, or in case of a PWM, frequency <0.05) are added to the list of candidate mutation sites, since they may be stability liabilities and represent good candidate sites to further optimize solubility and stability. The lower bound on the conservation index is necessary to avoid flagging positions that are intrinsically poorly conserved (e.g., within the CDR3 of an antibody). We refer to this group of candidate mutation sites as *identified from conservation*.

Fourth, those residues that are exposed on the surface, and that have PSSM-permitted mutations predicted to increase the solubility are also added to the list of candidate mutation sites, unless the position has a conservation index greater than 0.7 and the wild-type residue is already the consensus residue. This group of candidate mutation sites is referred to as *identified from exposed solubility*, as mutations here can be exploited to further increase the solubility and stability even if the sites are not predicted as liabilities. If the input protein has already got many candidate mutation sites in the previous three classes (maximum total number currently set to 100), then sites in this latest class are discarded from the list of candidates to safeguard computational efficiency.

The typical number of mutation sites in each of these four groups strongly depends on the protein under scrutiny. The final report from the webserver contains a table with all identified candidate mutation sites, which include information on how each site was identified (column "identified from", see Supplementary files 1–9).

Last, an option is provided for the user to input *custom* mutation sites, which, if given, are considered in the next steps of the pipeline alongside those identified automatically. *Custom* mutation sites can be exploited for example to remove known chemical or post-translational liabilities, such as deamidation or glycosylation sites, with mutations predicted to improve solubility and/or stability.

**Single mutational scanning.** Once the candidate mutation sites are identified, all possible amino acid substitutions allowed by the PSSM are tested as candidate mutations at each site. The user can decide which criterion to use for allowed substitutions from the filtering in Fig. 1B. The two options are either all substitutions with log-likelihood >0 (i.e., observed more than random, positively enriched), or only

those that also increase the frequency from the WT residues (log-likelihood > 0 & Δlog-likelihood >0), which is the default. Based on the results in Fig. 1B, the first option should be used in those cases where too few or no mutations are suggested by the algorithm at the end of the pipeline when running with the default log-likelihood >0 & Δlog-likelihood >0.

Allowed substitutions at each site are then singly ranked using the CamSol intrinsic solubility score, as this calculation is extremely fast[4]. Except for mutations happening at non-solvent-exposed sites identified from conservation, all mutations predicted to decrease solubility are discarded at this step. The output of this first part of the algorithm thus consists of a longlist of all PSSM-permitted mutations at all identified sites, comprising only solubility-improving mutations at those sites identified based on their low solubility.

This longlist is then used as the starting point for the calculation of stability change upon mutation with the energy-function Fold-X[38]. The energy-function is used to calculate for each longlisted mutation a ΔΔG score. This is the predicted difference in stability (ΔG) between the wild type and the mutant[62]. Only mutations with calculated ΔΔG < 0 are predicted to be stabilizing (or at least not de-stabilizing), and consequently further considered in the method pipeline. Therefore, the shortlist of candidate mutations contains single-point mutants characterized by the difference in CamSol solubility score between mutant and WT (Δ CamSol score), the difference in calculated stability (ΔΔG), and the difference in frequency (Δlog-likelihood). We are thus left with a list of mutations that are singly predicted to increase protein solubility and/or conformational stability.

If identical chains were identified in the input processing, the effect of the shortlisted point mutations is propagated to all chains identical to the chains on which they had been modelled. This means that the CamSol score is re-calculated after mutating the sequence of identical chains, and that the ΔΔG, initially calculated on only one chain for computational efficiency, is multiplied by the number of chains in the same group (this approximation is then tested at step (v) if needed).

Finally, these scores reflecting the changes in solubility, stability, and conservation are combined into the Mutation Score (see Supplementary Methods), which is used to provide a preliminary ranking of the single mutations. An intermediate output table is produced, containing the results of the single-mutational scanning. Mutations are named with a single string concatenating WT amino acid, Chain ID, pdb residue number, and mutated amino acid (e.g., LA24D to denote a leucine to aspartic acid substitution at residue number 24 of chain A). The final report from the webserver contains an extract of this table with those substitutions that improve the Mutation Score, and scatter plots showing predicted solubility and stability gains (see Supplementary files 1–9). The full table with all explored substitutions is also provided by the webserver as a.csv file inside the output zip folder. Therein, users can find all details of each attempted mutation, including its calculated contributions to the FoldX total energy (e.g., electrostatics, hydrogen-bonds, solvation, etc.).

**Combining multiple mutations.** The shortlisted mutations are then combined to create designs harbouring multiple mutations. Before beginning the combination, the single-point mutants with a negative Mutation Score (if any) are discarded. Then, all Δlog-likelihood of the single mutations are normalized, dividing by the standard deviation of the PSSM, to make them comparable among different chains.

To increase computational efficiency, the mutation combination process does not re-run all calculations for each possible combination of mutations, as in particular FoldX requires significant processing time. In the first instance, point mutations are combined by summing their ΔΔGs and normalised Δlog-likelihoods, while the intrinsic solubility score is re-calculated from the sequences harbouring all mutations, as the CamSol intrinsic calculation is adequately fast

(~200 sequences/seconds on a single core[23]). A combination of mutations is flagged as "potentially interacting" if at least two of its mutation sites are in proximity in the input structure, as assessed from the contact map that was calculated at the beginning in the CamSol structurally corrected solubility calculation[37] (see Supplementary Methods). The underlying assumption for this procedure is that mutations that are distant in the structure may be expected to yield an addictive contribution to the stability, or in other words that the overall stability change can be calculated by summing the stability changes of the individual mutations. However, this assumption breaks down when the mutations are close to each other in the structure. Therefore, combinations containing at least two mutations in proximity are flagged so that, if they occur in the final shortlist, their predicted stability can be recalculated by explicitly modelling these combinations (see step (v)).

Once the three metrics (ΔCamSol score, FoldX ΔΔG, and Δlog-likelihood) are computed for all combinations, the Mutation Score is calculated. If multiple identical chains are present in the input structure, the CamSol and ΔΔG scores are re-calculated as explained in section (iii). Point mutations are combined recursively in this way until the maximum number of simultaneous mutations decided by the user is reached, or until all identified suitable mutation sites have been combined, whichever happens first.

To preserve computational efficiency, the combinatorial space is gradually reduced during the combination process. Starting from the triple combinations onward, the algorithm considers only the top-ranking substitution for each candidate mutation site from the single-mutational scanning, as ranked by their Mutation Score.

We define a group of combinations as the ensemble of all combinations with the same total number of mutations (e.g., double mutants, triple mutants, etc.). Within each group, combinations are ranked according to their Mutation Score. Because of the addictive nature of the mutation score, the predicted top-ranking combination across all groups is almost certainly in the group with the maximum number of mutations attempted. However, this may not be beneficial in practice, as with each new mutation the chances of introducing a false positive in the combination increase. Therefore, a procedure is implemented to select those groups that embody the best balance between the number of mutations and predicted gain in solubility and/or stability.

The best groups are identified by those points in the recursive combination process, where the gain of carrying out an additional mutation becomes less favourable than it has been for the preceding mutations, as assessed by the growth of the Mutation Score as a function of the number of mutations (see Supplementary Methods). This procedure thus identifies one or more groups that embody the maximum gain with the minimum number of mutations (Fig. S4). Then, the algorithm creates a final shortlist of designed variants, which contains the three top-ranking combinations for each of these best groups, as well as the top-ranking combination from all other groups, so that at least one combination per total number of mutations is proposed in the final output for the user to consider.

**Check for potentially interacting mutations.** If any design flagged as "potentially interacting" ends up in the final shortlist, then its mutations are explicitly modelled one by one to make sure that the stability of the variant is not compromised by unfavourable interactions between different mutations. This process enables to test the assumption that the ΔΔG for a combination can be calculated as the sum of the ΔΔGs of its single-point mutations. Operationally, this is achieved by carrying out each mutation in the combination under scrutiny subsequently, using the output model of the previous step as input, so that the final model contains all mutations.

All mutations considered up to now where singly predicted to be stabilising (ΔΔG < 0). Therefore, if this test discovers a mutation predicted to be destabilizing (ΔΔG ≥ 0), it indicates unfavourable interactions with other mutations that have already been modelled at nearby positions. If such mutation is found, the algorithm tries to replace it with another substitution among those that were shortlisted for the same position by the single mutational scanning step. If applying this replacement results in a ΔΔG < 0, then the alternative mutation is kept. If this is not the case, the process is repeated to explore up to three alternative substitutions per position. If no suitable alternative is found, the disruptive mutation site is removed from the combination under scrutiny.

After this check, the Mutation Score of the combination is updated with the new ΔΔG that has been explicitly calculated. If the new score is lower than that initially predicted by summing the ΔΔG of the point mutations, a comparison is carried out with all other designs in the same group, as now a different combination with a higher Mutation Score might exist. If this is the case, the new top-ranking design is shortlisted, and the mutation-interaction check is repeated on it. If needed, this procedure is repeated up to three times for each group. Ultimately, the best ranking design among those combinations that have been checked (or one without the "potentially interacting" flag) is returned as the best for its group. This process, therefore, updates the final shortlist and the predicted best designs.

The final output consists of an html report with a description of the top-ranking designs, and the key results of each step of the pipeline, including graphs and descriptions (see for example Supplementary files 1–9). A zip folder is also provided containing detailed CSV tables with the results of all calculations, and the modelled structure of all top-ranking designs in PDB format.

## Experimental validation

After developing the pipeline, we first applied it to a nanobody that was isolated from a recently introduced naïve yeast-display library[63]. This nanobody, called Nb.b201, binds to human serum albumin with a KD in the high nanomolar range. The automated design procedure was run by allowing a maximum of 6 simultaneous mutations, with the phylogenetic filtering of log-likelihood > 0 only, and by using the precompiled MSA of single-domain antibody sequences (see Supplementary Methods). As input, we used the structure of Nb.b201 without the antigen, and paratope residues were excluded from the list of candidate mutation sites (chain C of PDB ID 5vnw, paratope residue numbers: 33,50,52,58,98,102-105).

The first step of the method is to compare the input sequence with the PSSM of the single-domain antibody MSA (Fig. 3A). Nb.b201 originated from a library based on the consensus sequence of nanobodies observed in the PDB[63]. Therefore, we find that the framework regions are mostly identical to the consensus sequence (top row of the matrix), thus focusing the candidate mutation sites to the CDR regions. The CamSol structurally corrected profile (Fig. 3B) reveals some small aggregation hotspots that cluster together on the surface (Fig. 3C), the larger of which comes from a paratope region in the CDR3, around residue F105. However, because the paratope regions were excluded from the design, the algorithm did not directly flag these sites for mutation. Conversely, using the criteria described in section "(ii) Selection of candidate mutation sites", the algorithm identified a total of 30 possible sites for mutation (Fig. 3C and Supplementary file 2), some of which are relatively close to the hotspots and may thus provide compensatory mutations. Of these 30 sites, 7 were shortlisted at the end of the single mutational scanning step, yielding a total of 17 different point mutations (Fig. 3D). The other sites were discarded because either no mutation was allowed by the PSSM at that position, or none of the allowed mutations was predicted to be solubilising by CamSol, or had a negative FoldX ΔΔG. All mutations at these seven sites were then combined into multiple designs harbouring up to 6 simultaneous mutations and, following the check for potentially interacting mutations, the best designs were returned (Fig. 3E).

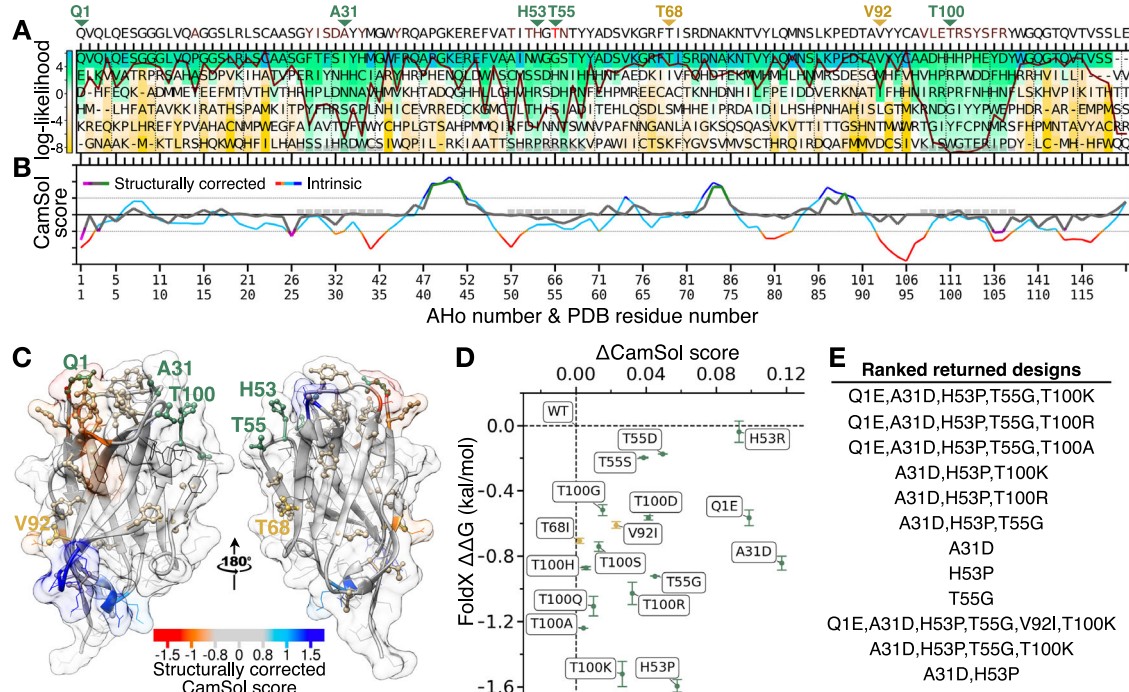

**Fig. 3 | Main pipeline steps for Nb.b201. A** Representation of the PSSM calculated from an alignment of 1396 single-domain antibody sequences. The log-likelihood at each position (colour-bar) is used to select candidate amino acid substitutions (allowed log-likelihood > 0). The sequence above the panels is the wild-type Nb.b201, residues in red have log-likelihood <0 (T55). The dark red line is the conservation index (high means position more conserved). Grey boxes denote CDR positions. Green or gold triangles point to mutation sites where at least one candidate mutation was shortlisted by the single-mutational scanning step (panel D). The last two residues (LE) do not have PSSM information, as they belong to a cloning restriction site. **B** CamSol profiles of Nb.b201. The CamSol intrinsic profile is coloured red to blue, where red means aggregation-prone and blue aggregation-resistant. It is common for globular proteins to have large aggregation-prone regions in their intrinsic profile that drive the hydrophobic collapse during folding.

The structurally corrected profile is coloured in grey/green/magenta, regions of low negative scores (magenta) are potential aggregation hotspots, regions of high score (green) are solubility promoting. **C** The CamSol structurally corrected profile is colour-coded (see colour-bar) on the surface of Nb.b201. Sidechains shown as "ball & stick" are identified candidate mutation sites, those labelled in gold or green are sites where at least one mutation was shortlisted at the single mutational scanning step, those labelled in green are also found among the top-ranking designs returned at the end. Figure made with UCSF Chimera. **D** Results of the single mutational scanning step as a scatter plot of ΔCamSol score (x-axis) against FoldX ΔΔG (y-axis, data points are means and error bars are SD over 3 runs) for all shortlisted mutations (ΔCamSol>0 & ΔΔG < 0). **E** Final designs returned by the pipeline, ranked as described in the main text.

Based on the analysis, we set out to produce Nb.b201 variants with the proposed mutations to evaluate their effect on both stability and solubility (see Supplementary Methods). We decided to test seven different single mutational variants, so to singly cover more of the mutations predicted to be beneficial, three double mutants, and the best predicted triple and quadruple mutants (Table 1). Although the top-ranking designs contained five mutations (Fig. 3E), we excluded Q1E from experimental testing, as the nanobody variants were produced in HEK293 cells, which yields N-terminal pyroglutaminated glutamine (see Table S2 and Supplementary Methods). Therefore, had we tested this mutation, we would have measured the difference between pyroglutamine and glutamic acid, instead of the predicted difference between glutamine and glutamic acid. We note, however, that the mutation Q1E is well-known in the nanobody field, and it was previously shown to be stabilising or neutral in a highly diverse set of eight nanobodies (mean sequence identity = 0.67)[64], thus suggesting that this prediction is unlikely to be wrong.

All nanobody variants were obtained at high purity (Table S2). The circular dichroism (CD) spectra of triple and quadruple mutants, as well as those of the single mutants not contained in the quadruple mutant, were indistinguishable from that of the WT and fully compatible with a well-folded VHH domain (Fig. S5). A biolayer interferometry (BLI) experiment confirmed that all variants bound their antigen (HSA) with $K_D$ values in the high nanomolar range in agreement with previous reports for Nb.b201[63] (Fig. S9, Figs. S13–17 and Table S5).

Conformational stability was measured with heat denaturation using nano differential scanning fluorimetry (nanoDSF, see Supplementary Methods). Strikingly, all variants had an apparent melting temperature greater than that of the WT (Fig. 4 and Table 1). The most stable variant was the quadruple mutant, with an apparent melting temperature greater than that of the WT by 13.6 °C.

We then attempted to measure relative solubility with polyethylene glycol (PEG) precipitation, using a recently introduced method[65]. However, a first experiment carried out with WT and quadruple mutant revealed that these nanobodies do not precipitate in PEG-6K at PEG concentrations up to 30% (weight/volume). A mild drop in soluble concentration was observed only for the WT at 33% PEG (Fig. S6), a concentration at which the pipetting of the automated robot starts to become less accurate because of the very high viscosity of the PEG solution[65]. While this result hints at a higher relative solubility of the quadruple mutant, it shows that PEG precipitation cannot be used to measure the relative solubility of all designed variants. Therefore, we resorted to using ammonium sulfate (AMS) instead of PEG as a precipitant in the assay. Although the two precipitants work through different principles, good correlations between these two types of protein precipitation measurements are reported in the literature[66], and AMS has previously been used to assess the relative solubility of monoclonal antibodies[14]. Our results reveal that all designed variants had a midpoint of AMS precipitation greater than that of the WT (Fig. 4A and S7). The assay, however, was not accurate enough to determine the rank-order of all variants with certainty, and

the confidence intervals of the fitted $AMS_{50\%}$ are typically very broad (Fig. 4A, S7 and Table 1).

To get more accurate estimates, we carried out measurements of cross interaction chromatography (CIC, Supplementary Methods)[67,68]. Higher retention times (RTs) in this chromatography technique indicate a higher degree of non-specific interactions with the IgG mixture (IgG pool purified from human serum) immobilised on the resin, or increased 'stickiness'. Owing to the strong correlation observed with solubility measurements, CIC was originally proposed as an assay to identify highly soluble antibody candidates[67]. A similar correlation with solubility was also reported more recently for a library of 17 monoclonal antibodies[14]. We find that, unlike AMS-precipitation measurements, CIC measurements were highly accurate, with duplicate experiments yielding virtually identical RTs (Fig. 4B).

Our results reveal that 8 designs, including triple and quadruple mutants, had better CIC performance than the WT, showing up to 49 seconds decrease in RT (Fig. 4B). Two variants, H53P and H53P T55G, showed similar performance to the WT, and two others, H53R and T100R, showed worse performance by 10.1 and 4.1 seconds, respectively. Notably, these two variants with higher CIC RTs than that of the WT are both mutations to arginine. Arginine residues in antibody CDRs have been associated with reduced specificity by several investigations[60,69–71]. Therefore, it is perhaps not surprising that these arginine variants perform worse in cross-interaction assays than they do in relative solubility assays such as the AMS precipitation.

Altogether, the results of this validation show that all designed Nb.b201 variants tested experimentally have improved conformational stability and relative solubility, and the majority of these designs also reduced cross-interactions. We also observed statistically significant correlations between the in-silico predictions underpinning variant selection in our pipeline (FoldX and CamSol) and the corresponding experimental measurements (Fig. S8).

To demonstrate the generality of these results, the computational method was run on two additional nanobodies and three antibody variable regions (Fv), comprising both VH and VL domains. We selected two anti-SARS CoV-2-RBD nanobodies, called H11-H4 and H11-D4, which were obtained by screening a llama-derived naïve library and further optimised via in vitro affinity maturation[72]. We also selected the CR3022 antibody, which originally was obtained from a SARS CoV-1 patient, but is also able to neutralise SARS CoV-2 and some of its variants of concern[73,74]. Finally, we selected the clinically approved antibodies adalimumab and golimumab (Humira® and Simponi®, respectively). These two antibodies both target human TNF-α, and were selected based on a study that characterised the biophysical properties of all clinical-stage antibodies, where adalimumab had no self-association or cross-reactivity flags, while golimumab had several[17]. The latter three antibodies were formatted as scFvs in this study.

For all the aforementioned antibodies, the antigen-bound structure was used as input for the automated design and the most stringent PSSM filtering was applied (log-likelihood & Δlog-likelihood > 0, see Supplementary Methods for more details). This approach differs from the one applied to Nb.b201, where the input was the structure of the nanobody without the antigen, all paratope residues were explicitly excluded from the design, and we used the looser PSSM filtering (log-likelihood > 0 only). We limited the design to a maximum number of mutations of 5 and 3 for scFvs and nanobodies, respectively (see Supplementary Methods for more details). In all cases, we selected the top design with the maximum number of mutations for experimental validation, in addition to several other designs for each antibody (Table S3).

As most mutations returned by the algorithm were within the CDR regions, we carried out an antigen-contact analysis on the bound WT structures to flag those mutations more likely to impact binding affinity. Three mutations were flagged in this way: L106P in H11-H4 and

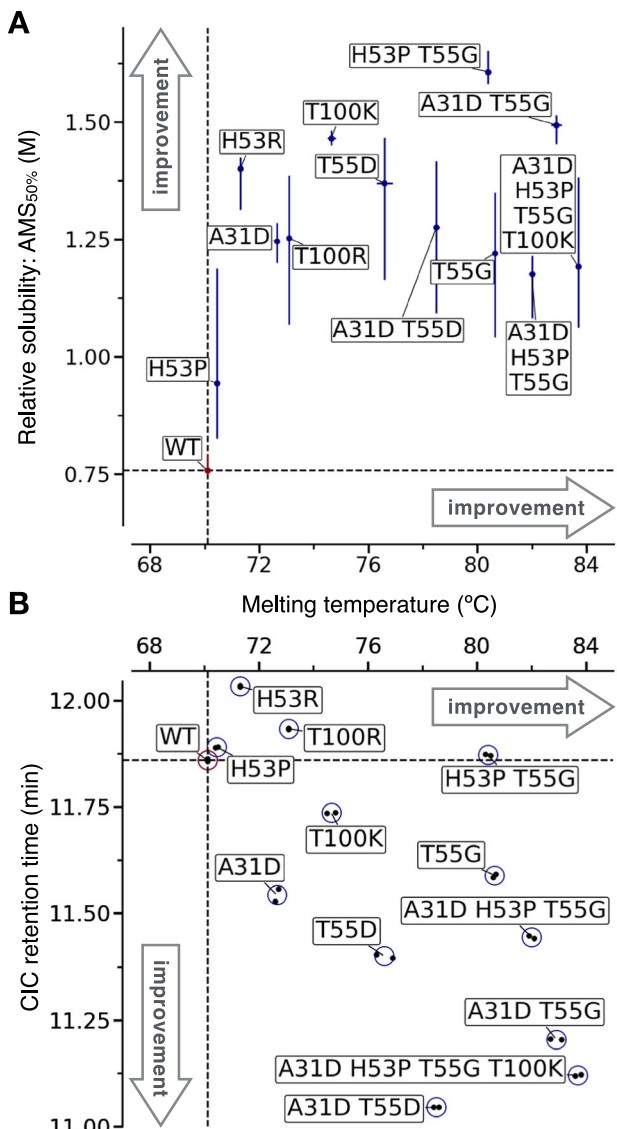

**Fig. 4 | Experimental characterisation of designed nanobody Nb.b201 variants. A** Scatter plot of the melting temperature ($T_m$) against the ammonium sulphate midpoint of precipitation ($AMS_{50\%}$, used as a proxy for relative solubility). Vertical error bars are the 95% confidence interval on the fitted $AMS_{50\%}$, horizontal error bars, which are often smaller than the data point, are standard deviation over two independent melting experiments (which are shown individually in **B**). **B** Scatter plot of the measured $T_m$ against the cross-interaction chromatography (CIC) retention time. Data from two independent experiments (black points) are reported. In both panels the dashed lines are drawn across the mean measurements of the WT Nb.b201 (red marker).

Y30G on the light chain of golimumab, both expected to disrupt binding, and T52S on the heavy chain of adalimumab, expected to be neutral (see Supplementary Methods). We decided to experimentally test these flagged mutations anyway, in the context of the returned top-ranking designs where they appear. However, in the case of the top H11-H4 and golimumab designs harbouring the two flagged mutations expected to disrupt binding, we also tested the corresponding designs obtained by excluding such mutations (Table S3).

In total 27 additional antibody variants were tested: the 5 WTs and 22 designs. We found that all variants had good purity and expected MW (Table S2). For each variant, we measured apparent melting temperature ($T_m$), cross-interaction chromatography retention time (CIC RT), and binding affinity (KD) using BLI.

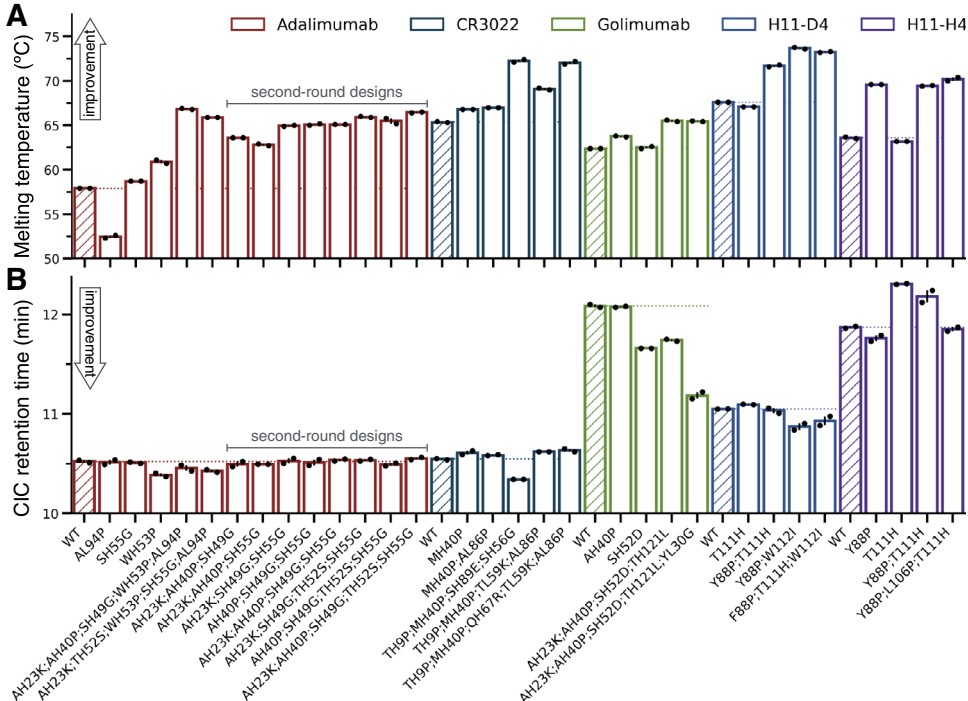

**Fig. 5 | Experimental characterisation of designs from five antibodies. A** Bar plot of the melting temperature ($T_m$) for each variant (x-axis labels). **B** Bar plot of the cross-interaction chromatography (CIC) retention time (RT). In both panels, data from two independent experiments (black points) are reported, and the height of the bar is their mean. Bars are coloured in groups according to the WT scFv or nanobody of each variant (see legend), and the bar corresponding to the WT is hatched. A coloured horizontal dotted line demarks the measurement of the WT in each group and serves as a guide to the eye.

Our results revealed that 19 of 22 designs (86.4%) had an improved $T_m$ over that of the corresponding WT (Fig. 5A), thus confirming the low false discovery rate of our automated design procedure. The largest improvements in stability for each antibody were $\Delta T_m = 9\,°C$ for adalimumab (5 mutations), 3.1 °C for golimumab (4 or 5 mutations), 6.9 °C for CR3022 (4 mutations), 5.7 °C for H11-D4 (3 mutations), and 6.6 °C for H11-H4 (3 mutations). Two of the three designs with worsened $T_m$ are H11 nanobodies harbouring the mutation T111H. H11-D4 and H11-H4 are closely related nanobodies that are affinity-matured variants of the same parental nanobody H11[72]. For both, the loss in $T_m$ is small ($\Delta T_m \geq -0.5\,°C$). Therefore, the only problematic false positive in these stability predictions is the A94P mutation in the light chain of adalimumab, which leads to a loss of 5.5 °C in $T_m$. Notably, this destabilising mutation is also present in the two designs with five mutations, which nevertheless have a $T_m$ 8 and 9 °C greater than that of the WT (respectively for H:A23K-T52S-W53P-S55G;L:AL94 and A23K-A40P-S49G-W53P;L:A94P). This finding suggests a strong compensatory effect from the other four mutations.

Similarly, CIC RT results show that all designs have a retention time comparable to, or better than that of their WT (Fig. 5B). The only exception is, again, the mutation T111H in the nanobody H11-H4. This mutation increases the RT by 26.1 s over that of the WT, and by 15.6 s in the context of the double mutant Y88P-T111H, while Y88P alone improves the RT in agreement with predictions ($\Delta RT = -6.6\,s$). Surprisingly, the same T111H mutation performed on the closely related H11-D4 is essentially neutral ($\Delta RT = 2.8\,s$), suggesting that the local physicochemical environment plays an important role.

The analysis of the CIC data further reveals that the scFv CR3022 and adalimumab WT already have excellent CIC performance, with a RT at the low end of the dynamic range of the assay (Fig. 5B). This finding was expected for adalimumab, as it aligns with previous reports[17]. Also expected, was the poor performance of golimumab WT, whose RT is 1 minute and 28 seconds worse than that of adalimumab WT. We find that all adalimumab and CR3022 designs have RTs roughly

comparable to those of their WT, in agreement with these being already at the low end of the dynamic range of the assay. Conversely, the golimumab designs improve the RT substantially, with −20.7 and −54 s for the quadruple and quintuple mutants, respectively. The best performing single-mutational variant of golimumab, SH52D, improves the RT by −25.5 s. The golimumab designs showed the biggest improvement in CIC RT and the most modest improvement in stability ($\Delta T_m = 3.1\,°C$), while the opposite is true for adalimumab designs ($\Delta RT = -6.6\,s$ and $\Delta T_m = 9\,°C$). This observation suggests that the automated design, when operating within the constraint of maximum five mutations, successfully prioritised mutations that primarily address the main liability of the antibody under scrutiny.

Finally, measurements of binding affinity at the BLI reveal that most designs have a KD comparable to that of the WT (Fig. S9). Notable exceptions, with a KD more than 3-fold worse than that of the WT, were designs harbouring the two mutations flagged prior to antibody production (L106P in H11-H4 and Y30G in golimumab), and all adalimumab designs harbouring the W53P mutation in the heavy chain. More specifically, as expected from the antigen-contact analysis, the proline substitution L106P in the CDR3 ablated the binding of H11-H4, while the Y30G substitution in the light chain of Golimumab increased the KD by a little more than 3 folds, in line with the removal of an antibody-antigen hydrogen bond (see Supplementary Methods).

While a loss of affinity is undesirable in antibody engineering, testing the flagged mutations helped to verify the predictions of the automated design. The L106P mutation leads to a $\Delta T_m$ of 0.8 °C and a $\Delta RT$ of −19.8 s, while the Y30G mutation leads to a $\Delta T_m$ of 0 °C and a $\Delta RT$ of −33.3 s (obtained comparing H11-H4 Y88P-L106P-T111H with H11-H4 Y88P-T111H and golimumab AH23K-AH40P-SH52D-TH121L-YL30G with golimumab AH23K-AH40P-SH52D-TH121L, respectively). The improved $\Delta RT$ is in line with the predictions, as these mutations singly contributed the biggest improvement in CamSol score when compared to all other suggested single mutations. Similarly, the small effect on $\Delta T_m$ reflects the modest FoldX $\Delta\Delta G$ of only −0.5 kcal/mol and

**Table 1 | Experimentally characterised designed Nb.b201 variants**

| Nb.b201 variant | $T_m$ (°C) | $AMS_{50\%}$ (M) | CIC RT (min) | $\Delta T_m$ (°C) | $\Delta RT$ (s) |
|---|---|---|---|---|---|
| WT | 70.1 ± 0.05 | 0.76 in (0.75,0.79) | 11.859 ± 0.003 | 0 | 0 |
| A31D | 72.7 ± 0.1 | 1.25 in (1.20,1.28) | 11.543 ± 0.014 | 2.6 ± 0.1 | −19.0 ± 1.1 |
| H53P | 70.5 ± 0.05 | 0.94 in (0.82,1.19) | 11.889 ± 0.001 | 0.4 ± 0.05 | 1.8 ± 0.1 |
| H53R | 71.3 ± 0.05 | 1.40 in (1.31,1.42) | 12.033 ± 0.002 | 1.2 ± 0.05 | 10.4 ± 0.3 |
| T55D | 76.6 ± 0.3 | 1.37 in (1.16,1.47) | 11.399 ± 0.003 | 6.5 ± 0.3 | −27.6 ± 0.0 |
| T55G | 80.7 ± 0.1 | 1.22 in (1.04,1.35) | 11.588 ± 0.003 | 10.6 ± 0.1 | −16.2 ± 0.0 |
| T100K | 74.7 ± 0.1 | 1.47 in (1.45,1.48) | 11.736 ± 0.001 | 4.6 ± 0.1 | −7.4 ± 0.1 |
| T100R | 73.1 ± 0.05 | 1.25 in (1.07,1.39) | 11.932 ± 0.002 | 3.0 ± 0.05 | 4.4 ± 0.3 |
| A31D T55D | 78.5 ± 0.1 | 1.27 in (1.09,1.42) | 11.045 ± 0.001 | 8.4 ± 0.1 | −48.8 ± 0.2 |
| A31D T55G | 82.9 ± 0.2 | 1.49 in (1.45,1.51) | 11.204 ± 0.001 | 12.8 ± 0.2 | −39.3 ± 0.2 |
| H53P T55G | 80.4 ± 0.1 | 1.61 in (1.58,1.65) | 11.872 ± 0.002 | 10.3 ± 0.1 | 0.8 ± 0.3 |
| A31D H53P T55G | 82.0 ± 0.1 | 1.18 in (1.08,1.22) | 11.444 ± 0.003 | 11.9 ± 0.1 | −24.9 ± 0.4 |
| A31D H53P T55G T100K | 83.7 ± 0.1 | 1.19 in (1.06,1.38) | 11.120 ± 0.001 | 13.6 ± 0.1 | −44.3 ± 0.2 |

WT and rationally designed mutational variants with their measured conformational stability (apparent melting temperature, $T_m$; apparent because heat denaturation is not reversible), relative solubility in ammonium sulphate precipitation ($AMS_{50\%}$), and cross interaction chromatography retention time (CIC RT). Values are mean ± standard deviation, except for the relative solubility column, which reports fitted $AMS_{50\%}$ and lower and upper 95% confidence interval on this fitting parameter (see Fig. S7). $T_m$ was measured in duplicates and reported at a 0.1 °C resolution, when the two measurements were identical a standard deviation of 0.05 was arbitrarily assigned. Δ are calculated as differences between the value of the variant under scrutiny and that of the corresponding WT.

−0.01 kcal/mol for L106P and YL30G, respectively (see Supplementary files 8 and 5, single mutational scanning section therein).

The WH53P mutation in adalimumab is improving stability and cross-reactivity in agreement with the predictions ($\Delta T_m = 3$ °C and $\Delta RT = -8.1$ s over the WT). However, it led to a substantial affinity loss even if it was not flagged as it's not in direct contact with the antigen (see Supplementary Methods). In hindsight, the substitution to proline is highly likely to affect the loop conformation and thus the binding affinity, especially as the adjacent residue at position 52 is in direct contact with the antigen. As the WH53P mutation is present in all adalimumab multi-mutant designs, and so is AL94P, which was unfavourable in regards to $T_m$, we decided to produce and evaluate a second round of adalimumab designs consisting of combinations of the other mutations (Table S3).

These second-round adalimumab designs comprised four triple mutants, three quadruple mutants, and one quintuple mutant. The latter contains all mutations returned by the automated design except for WH53P and AL94P. All 8 designs had a melting temperature greater than that of the WT, by a minimum of 4.9 °C and a maximum of 8.6 °C for the triple mutant AH23K-AH40P-SH55G and the quintuple mutant AH23K-AH40P-SH49G-TH52S-SH55G, respectively (Fig. 5A). All second-round designs had a CIC RT comparable to that of the WT, which is already at the low end of the dynamic range of the assay (Fig. 5B), and bound TNF-α with a KD at least as good as that of the WT (Figs. S9, S14 and Table S4).

As a further validation, we benchmarked the predictions of our method with published experimental data for generic proteins (not antibodies). Since stability and solubility are crucial properties of recombinant proteins, there are many reports of experimentally characterised mutations that improve at least one of these properties[62,75,76]. However, our method automatically identifies specific mutations that should be performed, which are thus unlikely to be those that were characterised in previous studies. To circumvent this issue, we looked at Deep Mutational Scanning (DMS) data[77]. In DMS, a library of variants is created and introduced into a system in which the genotype is linked to a selectable phenotype. When selection for the function of the protein is imposed, variants with high activity increase in frequency, whereas variants with low activity decrease in frequency. High-throughput DNA sequencing is used to measure the frequency before and after selection, and a fitness score reflecting the activity of each variant is calculated[77,78]. DMS libraries are usually constructed by mutating one site at the time, often to all 19 alternatives, across the full

length of the protein[77]. Therefore, fitness scores are typically available for most of the mutations predicted by our method, making these data particularly suitable for a benchmark. Nonetheless, a limitation of such benchmark is that DMS data report on the functionality of the protein, not on its stability and solubility. On one hand, variants with hindered stability or solubility should express poorly, and hence have low fitness scores. On the other, however, variants with the highest fitness scores are typically not the most stable or soluble, but those with the best activity. Consequently, mutations suggested by our method should have higher fitness than randomly performed mutations but would not necessarily be top-ranking.

We obtained DMS data[77] from the work of Kovilakam Sruthi et al.[78], who collated data for seven unrelated proteins from various sources. We ran the CamSol Combination pipeline on these proteins by allowing a maximum of one mutation, as fitness scores are only available for one mutation at a time (see Supplementary Methods). Sites for mutation are identified automatically by our method based on their conservation and contribution to solubility (section "(ii) Selection of candidate mutation sites"). We find that, for all seven proteins, mutations at these sites correspond to variants with higher fitness scores than the background distribution of mutations at all sites ($p$-value $\leq 10^{-5}$, Fig. S10, see Supplementary Methods). The method then carries out all PSSM-allowed mutations at the identified sites and shortlists those that are singly predicted to improve conformational stability and solubility (section "(iii) Single mutational scanning"). For all seven proteins, we find a very consistent trend, whereby the distribution of fitness scores of shortlisted mutations is more skewed toward high fitness than that of all DMS-characterised mutations at the same sites, which in turn is more skewed towards high fitness than that of all mutations at all sites (Fig. S10). This trend meets the threshold for statistical significance ($p < 0.05$) for four proteins, and for six when comparing fitness scores of shortlisted mutations to the background of all mutations (Fig. S10B). We note that p-values greater than 0.05 are associated with a small number of shortlisted mutations, rather than with a different trend in the fitness scores of shortlisted mutations (Fig. S10A, B). Overall, despite the limitation of benchmarking with DMS data reporting on protein function, we find that shortlisted mutations are associated with high experimental fitness. This finding suggests that the false discovery rate of our computational procedure is low also for generic proteins, as poorly expressing variants and mutations that disrupt protein function are not shortlisted.

**Table 2 | Experimentally characterised designed antibody variants**

| Antibody | Variant | T$_m$ (°C) | CIC RT (min) | ΔT$_m$ (°C) | ΔRT (s) |
|---|---|---|---|---|---|
| Adalimumab | WT | 57.9 ± 0.05 | 10.521 ± 0.011 | 0 | 0 |
| | AL94P | 52.5 ± 0.2 | 10.514 ± 0.022 | −5.4 ± 0.2 | −0.4 ± 2.0 |
| | SH55G | 58.7 ± 0.05 | 10.509 ± 0.008 | 0.8 ± 0.05 | −0.7 ± 0.2 |
| | WH53P | 60.9 ± 0.2 | 10.386 ± 0.018 | 3.0 ± 0.2 | −8.1 ± 0.5 |
| | AH23K AH40P SH49G WH53P AL94P | 66.8 ± 0.1 | 10.456 ± 0.028 | 9.0 ± 0.1 | −3.9 ± 1.1 |
| | AH23K TH52S WH53P SH55G AL94P | 65.9 ± 0.05 | 10.427 ± 0.014 | 8.0 ± 0.05 | −5.6 ± 0.2 |
| | AH23K AH40P SH49G | 63.6 ± 0.05 | 10.495 ± 0.025 | 5.7 ± 0.05 | −1.6 ± 2.2 |
| | AH23K AH40P SH55G | 62.8 ± 0.1 | 10.495 ± 0.000 | 4.9 ± 0.1 | −1.6 ± 0.6 |
| | AH23K SH49G SH55G | 65.0 ± 0.05 | 10.527 ± 0.026 | 7.1 ± 0.05 | 0.4 ± 2.2 |
| | AH40P SH49G SH55G | 65.1 ± 0.1 | 10.512 ± 0.029 | 7.2 ± 0.1 | −0.5 ± 2.4 |
| | AH23K AH40P SH49G SH55G | 65.1 ± 0.05 | 10.536 ± 0.010 | 7.2 ± 0.05 | 0.9 ± 1.2 |
| | AH23K SH49G TH52S SH55G | 66.0 ± 0.05 | 10.534 ± 0.009 | 8.1 ± 0.05 | 0.8 ± 1.2 |
| | AH40P SH49G TH52S SH55G | 65.5 ± 0.3 | 10.490 ± 0.014 | 7.6 ± 0.3 | −1.9 ± 1.5 |
| | **AH23K AH40P SH49G TH52S SH55G** | **66.5 ± 0.05** | **10.552 ± 0.010** | **8.6 ± 0.05** | **1.9 ± 1.3** |
| CR3022 | WT | 65.3 ± 0.1 | 10.545 ± 0.005 | 0 | 0 |
| | MH40P | 66.8 ± 0.05 | 10.610 ± 0.020 | 1.4 ± 0.1 | 3.9 ± 1.5 |
| | MH40P AL86P | 67.0 ± 0.05 | 10.585 ± 0.005 | 1.6 ± 0.1 | 2.4 ± 0.6 |
| | **TH9P MH40P SH89E SH56G** | **72.2 ± 0.2** | **10.340 ± 0.000** | **6.9 ± 0.2** | **−12.3 ± 0.3** |
| | TH9P MH40P TL59K AL86P | 69.1 ± 0.1 | 10.620 ± 0.000 | 3.8 ± 0.05 | 4.5 ± 0.3 |
| | TH9P MH40P QH67R TL59K AL86P | 72.1 ± 0.1 | 10.635 ± 0.015 | 6.7 ± 0.2 | 5.4 ± 0.6 |
| Golimumab | WT | 62.4 ± 0.05 | 12.085 ± 0.015 | 0 | 0 |
| | AH40P | 63.8 ± 0.05 | 12.075 ± 0.005 | 1.4 ± 0.05 | −0.6 ± 1.2 |
| | SH52D | 62.5 ± 0.1 | 11.660 ± 0.000 | 0.1 ± 0.1 | −25.5 ± 0.9 |
| | **AH23K AH40P SH52D TH121L** | **65.5 ± 0.1** | **11.740 ± 0.010** | **3.1 ± 0.1** | **−20.7 ± 0.3** |
| | AH23K AH40P SH52D TH121L YL30G | 65.5 ± 0.05 | 11.185 ± 0.035 | 3.1 ± 0.05 | −54.0 ± 3.0 |
| Nb H11-D4 | WT | 67.6 ± 0.05 | 11.049 ± 0.002 | 0 | 0 |
| | T111H | 67.1 ± 0.05 | 11.095 ± 0.002 | −0.5 ± 0.05 | 2.8 ± 0.2 |
| | Y88P T111H | 71.7 ± 0.1 | 11.033 ± 0.026 | 4.1 ± 0.1 | −1.0 ± 1.6 |
| | **Y88P W112I** | **73.7 ± 0.1** | **10.871 ± 0.034** | **6.1 ± 0.1** | **−10.7 ± 1.9** |
| | F88P T111H W112I | 73.2 ± 0.05 | 10.929 ± 0.046 | 5.7 ± 0.05 | −7.2 ± 2.7 |
| Nb H11-H4 | WT | 63.6 ± 0.1 | 11.870 ± 0.010 | 0 | 0 |
| | **Y88P** | **69.6 ± 0.05** | **11.760 ± 0.030** | **6.0 ± 0.1** | **−6.6 ± 1.2** |
| | T111H | 63.2 ± 0.05 | 12.305 ± 0.005 | −0.4 ± 0.1 | 26.1 ± 0.3 |
| | Y88P T111H | 69.5 ± 0.05 | 12.180 ± 0.060 | 5.9 ± 0.1 | 18.6 ± 3.0 |
| | Y88P L106P T111H | 70.2 ± 0.2 | 11.850 ± 0.020 | 6.6 ± 0.3 | −1.2 ± 0.6 |

WT and rationally designed mutational variants with their measured apparent melting temperature (T$_m$), and cross interaction chromatography retention time (CIC RT). Values are mean ± standard deviation, T$_m$ was measured in duplicates and reported at a 0.1 °C resolution, when the two measurements were identical a standard deviation of 0.05 was arbitrarily assigned. Δ are calculated as differences between the value of the variant under scrutiny and that of the corresponding WT. The designs in bold embody the best balance of T$_m$, CIC RT, and binding KD (see Fig. S9).

In summary, we validated the method predictions on six different antibody fragments, including two approved therapeutic antibody and three nanobodies, and we benchmarked them against DMS data from seven unrelated proteins. We tested experimentally a total of 34 designs, plus 8 second-round adalimumab designs. We covered 33 different amino acid substitutions, some of which were tested singly and others only as part of combinations (Table S3). Most designs had improved stability and cross-reactivity, and we only identified two problematic false positive substitutions (AL94P in adalimumab and T111H in H11-H4). Even if most mutations are in the CDR region, only one unexpectedly affected the affinity (WH53P in adalimumab), and this was readily fixed by expressing a second round of designs based on the same computational predictions. The best designs for each antibody are reported in bold in Table 2. Taken together, these results demonstrate that our automated computational pipeline is highly effective at designing antibody variants with improved developability potential.

## Discussion

In this study, we have presented a fully automated computational pipeline and associated webserver for the design of conformationally stable and soluble protein variants, with a particular focus on immunoglobulin variable domains.

The simultaneous improvement of stability and solubility is of high relevance, as these two properties underpin many liabilities that can hamper the development of biologics, such as aggregation, low expression yield, or instability upon long-term storage. Furthermore, very high stability and solubility are required to access new administration routes for biologics, such as oral and inhalation delivery.

Our approach combines predictions of these two biophysical properties with the analysis of phylogenetic information. We have shown that including phylogenetic information significantly reduces the false discovery rate of stability predictions by analysing a large database of experimental data (Fig. 1). Furthermore, of the 34 designs from 6 different antibodies we tested experimentally, 31 (91%) had an

apparent melting temperature greater than that of their WT, as well as all 8 second-round adalimumab designs, confirming the low FDR of our stability predictions (Figs. 4 and 5). Other computational pipelines that combine aggregation propensity and stability predictions include Aggrescan3D v.2[36] and SolubiS[35]. However, these methods do not consider phylogenetic information, and only suggest single mutations to 'gatekeeper' residues (typically charged residues). Similarly, the PROSS[18,79] and the FireProt[19,80] webservers have been successful in designing mutations that improve stability, but do not explicitly consider aggregation propensity, model combinations of mutations on only one chain at a time, and their homolog search strategy is not readily applicable to immunoglobulin variable domains.

Our algorithm is agnostic of the protein structure and can handle input proteins comprising multiple polypeptide chains, takes into consideration residues with missing coordinates in its solubility predictions (using the *seqres* sequence), and handles immunoglobulin variable domains using custom-built precompiled MSAs. The pipeline is slightly different depending on whether the input is an immunoglobulin variable region or a generic protein. For a generic protein like an enzyme, a MSA is constructed from homologs found with the HHblits algorithm. In this case, the inclusion of phylogenetic information also safeguards against predicting mutations at functionally relevant sites, as these are typically conserved by evolution. Conversely, different pre-compiled MSAs are provided for immunoglobulin variable domains, representing the mutational space observed in human antibodies, mouse antibodies, nanobodies, and post-phase-I clinical-stage antibodies.

Functionally relevant antibody residues (i.e., the paratope) are found within hypervariable regions and therefore cannot be inferred from phylogenetic information. However, when bound structures or mutational studies are not available to determine paratope residues, paratope predictors can be used. These algorithms predict those residues most likely to be directly involved in antigen binding from the antibody sequence. Many such predictors are available and their accuracy has been rapidly increasing[81-85], and one, called Parapred[83], can be accessed directly on our webserver. Whether predicted or experimentally determined, functionally relevant positions can be excluded from the design, as we have done for Nb.b201 in this work.

The computational pipeline is fully automated and accessible via a user-friendly webserver. The input page enables the customisation of various parameters and settings, thus making the design highly tuneable to accommodate user-specific needs. Users can decide on target residues to exclude, sites or whole chains that should not be touched, can input extra "custom" mutation sites to test, choose between different PSSM thresholding and pre-compiled antibody MSA, and even upload user-built MSAs. The webserver implementation of the method is complemented by an easy-to-use graphical interface that guides the user, including a simple guide on how to generate and download suitable alignments of homolog sequences from the HHblits webserver[58].

The required input is the structure of the protein to be optimised. While an experimentally determined structure may not always be available, great advances are being made in the de novo modelling of proteins[86,87] and unbound antibody structures[88-91]. Models generated with such software, or simpler homology models, can readily be used as input for our algorithm. In the case of antibodies, we have run our pipeline on an unbiased set of 19 Fv regions modelled with the ImmuneBuilder webserver[92], and compared these results with those from runs on the corresponding crystal structures (see Supplementary Methods). We find that there is a strong agreement between predictions carried out on models and on corresponding structures (Fig. S11). In particular, it is rare for mutations present in any of the final designs obtained from a model to be predicted to be non-beneficial on the corresponding structure. Only 4 of 19 assessed Fv models contained such potentially problematic mutations in their final designs, and, in all four cases, it was only one mutation (Fig. S11). Overall, as approaches to

model protein and antibody structures are improving at an unprecedented rate, we expect that there will soon be no difference between running our pipeline on a crystal structure or on a model.

We experimentally validated the predictions of the algorithm on six antibodies: three nanobodies and three human antibodies, which we expressed as scFv. Overall, we produced 48 antibody variants, consisting of the 6 WTs, 34 designed mutational variants, and 8 second-round adalimumab designs. We measured thermal stability ($T_m$), cross interaction chromatography retention time (CIC RT), and binding affinity (KD) for all of these variants, and for variants of Nb.b201 also the midpoints of AMS precipitation. These are widely used in vitro developability assays, whose measurements are well-known to be predictive of solubility and of high concentration behaviour[43,65,67,93,94]. The key advantage of these assays, which is the reasons why they are so widely used, is that they require low amounts of purified protein material. To confirm the estimates of conformational stability obtained from the apparent melting temperatures ($T_m$), we also measured the ΔG from chemical denaturation of at least two variants per antibody (Fig. S12). We find that, even if ΔG measurements are affected by larger experimental uncertainties than $T_m$ measurements, there is a perfect agreement in the mutational variants' stability rankings obtained with chemical and heat denaturation (Table S4).

The results of these experiments demonstrate that the computational procedure is remarkably successful at designing antibody variants with increased stability and relative solubility, and reduced cross-reactivity. We only found two false positive mutations (T111H in the H11 Nbs, and AL94P in adalimumab), and two others (H53R and T100R in Nb.b201) that improved thermal stability and AMS precipitation midpoint, but worsened the CIC retention time.

We noted that the two Nb.b201 variants with worsened CIC retention times, but increased AMS precipitation midpoint, were both mutations to arginine in the CDRs, which are well-known to contribute to poor specificity[60,69-71]. This type of liability is easy to spot computationally, as most software for molecular viewing can highlight patches of positive charges. These can also be identified directly from the CDR sequences, and are tabulated in various publications[69,71]. Automated assessment of sequence-based liabilities and their removal will be the first area of improvement for future versions of the method, including known drivers of cross-reactivity (e.g., number of CDR arginine & tryptophan residues) as well as chemical liabilities (e.g., deamidation sites or post-translational-modification sites). At present, users have the option to exclude specific residues from the list of potential substitution targets. So, if any of the final designs contains liabilities, the calculation can be re-run by excluding the relevant residue(s) to get a different set of top-ranking designs (e.g., exclude arginine for specificity, asparagine for deamidation and glycosylation, etc.). Similarly, if any such liability is present in the WT protein, users can add the corresponding site as custom mutation site in the algorithm, so that mutations predicted to improve stability and solubility will be suggested to remove the liability.

In conclusion, we have introduced and experimentally validated a fully automated computational method that provides a time- and cost-effective way to improve the stability and solubility of proteins and antibodies, through the rational design of mutations. We anticipate that this algorithm will find broad applicability in the optimisation of the developability potential of lead proteins and antibodies destined to applications in research, biotechnology, diagnostics, and therapeutics.

## Methods
All materials and methods are available in the Supplementary Materials.

### Reporting summary
Further information on research design is available in the Nature Portfolio Reporting Summary linked to this article.

## Data availability

All data needed to evaluate the conclusions in this article, or that are necessary to interpret, verify and extend the research in the article are present in the paper and/or the Supplementary Materials and Supplementary files. Additional details are available from the corresponding author on request. Source data are provided with this paper.

## Code availability

The method is made available to the academic community as a webserver at www.cohsoftware.ch.cam.ac.uk/index.php/camsolcombination. In order to access the software, users need to register a free account and log in.

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

## Acknowledgements

P.S. is a Royal Society University Research Fellow (URF\R1\201461). This work was partly funded by a Research Grant (RGS\R1\211126) from the Royal Society, by an Isaac Newton Trust/Wellcome Trust ISSF/University of Cambridge Joint Research Grant (MBAG/624 RG89305), and by UKRI EPSRC (EP/X024733/1). Biomolecular production and some of the characterisations were funded by Novo Nordisk. M.O. is a Ph.D. student funded by AstraZeneca. A.B. and M.M.M are Ph.D. students within the Novo Nordisk R&D STAR Fellowship programme and are partially funded by Innovation Fund Denmark.

## Author contributions

A.R. and P.S. designed and coded the algorithm and carried out the computational analysis. A.B., M.O., L.S., C.P., N.L., and P.S. designed experiments. A.B., M.O., M.M.M., L.S., and N.L. carried out experiments. P.S. conceived and supervised the project. A.R., A.B. and P.S. wrote the first draft of the paper. All authors analysed data and edited the paper.

## Competing interests

A.B. and M.M.M. are industrial Ph.D. students at Novo Nordisk. C.P., L.S., and N.L. are employees of Novo Nordisk and may hold stocks or stock options. P.S. is one of the developers of the original CamSol method, which is available as a free web server, but also through commercial licenses. The remaining authors declare no competing interests.
