## [Peer Review File · Nature Communications]

Automated optimisation of solubility and conformational stability of antibodies and proteinsReviewer #1 (Remarks to the Author):

This is an interesting manuscript describing a computational pipeline for simultaneously improving folded state stability and solubility of proteins. Importantly, the effectiveness of the pipeline is demonstrated experimentally with different antibody and fragment structures. It is therefore valid for the authors to claim that the method has the potential for impact in biopharmaceutical, and other, applications.

The pipeline largely builds on existing methodology for solubility (CamSol) and stability (Fold-X) predictions, but confirming their applicability when computational and experimental values are compared. There is novelty in the combination of these methods with multiple sequence alignment data (for evolutionary suitability of mutations), and some tweaks of how to put it together. The pipeline is delivered via a web server.

I agree with the authors that the work is a potentially valuable tool for areas of biopharms and biotech that seek (and likely to be common) improvements in conformational and colloidal stability at the same time.

I feel that the manuscript is rather wordy, but I'm not clear how otherwise it could be constructed to improve the message, perhaps more detail into Suppl Info. Overall it is clear, and the figures good.

Some questions follow.

1. With regard to biopharmaceuticals, the authors mention the driver for increased stability at high concentrations, for storage and administration. Could they discuss the scale of improvement (e.g. mg/ml concentration) that their redesigns could typically achieve?
2. Early in the manuscript, a class of sites labelled as near surface (and possibly involved in aggregation) are noted. How many such sites make it into the final designs, versus the surface sites?
3. In the nanobody example, mutation to arginine is seen to be problematic which, as the authors note is a known phenomenon. How would a constraint to avoid this be added to the pipeline / should it?
4. For Fv example(s), there is a step of checking for interaction of a designed mutation with the antigen binding site, is this part of the pipeline or an add-on. If an add-on, how would it be put into the pipeline?
5. Do we get sufficient feel for what the mutations are doing, at least in terms of Fold-X $\Delta\Delta G$? e.g. for the nanobody there look to be several charge changes, are these just improving plus/minus network (and then solubility of course too)? Whereas in the Fvs there seem to be more non-charged mutations - what are the anticipated basis of stability change? Is it easy to design proline changes - do these come from Fold-X and MSA together?

Reviewer #2 (Remarks to the Author):

The authors report incremental improvements to an existing software for predicting mutations that improve protein solubility by incorporating a phylogenetic filter that identifies residues that are conserved amongst homologous proteins. By combining this with force field calculations that score stability, the resulting pipeline is expected to enable users to design proteins which have reduced propensity to aggregate. The authors have tested this by applying the algorithm to the sequences of three antibodies in scFvs format (VH-linker-VL) and three single domain nanobodies (VHH). The mutations predicted by the silico analyses have been comprehensively tested experimentally and show that the introduced changes increase thermal stability and solubility in some cases without compromising antigen binding activity. Given these use cases, the work is principally directed at industry users for increasing properties that favour the developability of antibodies as biopharmaceuticals. Although solubility/stability are key parameters, other considerations such as chemical modifications (e.g. deamidation and glycation) are also important in selecting antibody development candidates. The authors indicate that incorporating screening for such sequence-based liabilities forms part of future developments. Although the output of the pipeline provides choices, the experimental scientist using this pipeline still needs to make selections based on other

knowledge and experience.

Specific comments

1. Antibodies are highly sensitive to concentration and are often required to be formulated at tens of mg/ml. The experimental work reported was carried out at protein concentrations 0.1-1.0 mg/ml. This raises the question of how valid the predictions are for higher antibody concentrations.

2. There are several other programmes that have been developed to address the developability of antibodies, recently reviewed in Akbar et al 2022 (DOI: 10.1080/19420862.2021.2008790). Although in a different context, the use of a phylogenetic filter in the prediction of mutations in proteins that will improve solubility has also been reported (DOI:10.1016/j.molcel.2016.06.012 DOI 10.1093/bioinformatics/btaa1071). It would be appropriate to comment on the relative merits of different but similar pipelines.

3. The title of the article indicates the applicability of the pipeline to proteins in general but only one example is provided and without any experimental validation. It is suggested that either, further examples are included together with at least an experimental study or that the title is made more specific.

4. The choice of nanobodies in the test set is interesting but given that these are typically very soluble and stable proteins, the need to improve this by mutagenesis is unlikely to be a widely adopted application of the software.

5. The run time for the software is admirably fast but log files when runs fail would benefit from some explanation for the non-specialist user.

Reviewer #3 (Remarks to the Author):

The manuscript by Rosace, Bennett et al. presents CamSol combination – a computational pipeline to co-optimize solubility and thermodynamic stability of antibodies (and potentially other proteins) by predicting combinations of favorable mutations. Computational pipelines like this one exist and are of broad interest for rational optimization of antibodies. Two of the essential tools (i.e., CamSol and FoldX) used in the presented approach were published previously and have proven to yield reasonable predictions. To me, the novelty of CamSol combinations comes from the input of phylogenetic information to increase the accuracy of the predictions. The prediction of combinations of mutations is also a step forward. The authors demonstrate with several proteins and dozens of mutants that most of the computational predictions are reasonable. The paper is well written, the method is available via a website which I found user-friendly. The data output from the server is very clear. In my opinion, better validation and some additional explanations of the method will be highly beneficial for the manuscript and will increase the value to justify publication in this journal. I listed some comments (not in the order of importance):

- Studies on similar approaches combining colloidal and conformation stability predictions, e.g., TANGO and FoldX (PMID: 2832291), Aggrescan 3D and FoldX (PMID: 31049593), have to be mentioned and the advantages of CamSol combination over these methods should be clearly explained in the discussion. Would the automated Aggrescan 3D/FoldX prediction of mutations overlap to some extent with the individual mutations from CamSol combination?
- When using different databases (e.g., clinical-stage, OAE, abYsis) for the phylogenetic analysis, some of the predicted mutations overlap, but others do not. Non-expert users of CamSol combination will be faced with the challenge of deciding which databases to use. A better discussion (or maybe a table) highlighting the strong and weak points of using the different databases will be very useful. Comparisons and experimental validations for several antibody mutants optimized via CamSol combination using different sequence databases will be valuable.

- FoldX is known to work best with crystal structures as an input. However, the CamSol combination will most likely be used on homology models of antibodies. Can the authors show an unbiased comparison of the quality of the predictions when using crystal structures versus homology models? Does the server predict the same mutations?
- The melting temperature was used as a proxy for the conformational stability, but measurement of the ΔG of unfolding to validate the predictions would be more accurate. Can the authors provide ΔG values on selected mutants and the corresponding wild-types?
- It is not surprising that the ammonium sulphate precipitation did not yield reliable results to validate the predictions of the nanobodies. The cross-interaction chromatography might show some correlations with the predictions, but CIC is not really a measurement for colloidal stability. Can you use a light scattering approach to measure the second osmotic virial coefficient which is the actual thermodynamic parameter that should correlate with colloidal stability? Alternatively, affinity-capture self-interaction spectroscopy is a good proxy for colloidal stability.
- In the structures that I optimized with CamSol combinations, there were several interesting predictions. It appears that the colloidal stability of antibodies with basic isoelectric points is often increased by introducing more positive charges (which are known to drive polyspecificity). I did not observe a case where the method chose to optimize an antibody by introducing negative charges to significantly reduce the isoelectric point (although it is known that antibodies with acidic isoelectric points can exhibit high solubility and stability). What is the possible reason for this?
- A recent paper presented an in vitro approach to select stabilizing mutations in antibodies (PMID: 32286330). I wonder if the computational optimization with CamSol combination will predict some of the mutations reported in this paper.
- The validations for IgGs were performed by measurements on the scFv. Reformatting an scFv into a Fab can have unexpected effects on protein solubility/stability. Can the authors validate their predictions for several mutants produced as full-length IgGs?
- It will be highly valuable if the authors show correlations between the CamSol combination predictions and the production yields of the proteins.
- There is no validation of the method with proteins different than antibodies, although the title implies that the approach also works with other proteins. There is plenty of data on the effect of mutations on enzymatic stability/solubility (except for the used ProTherm database). The authors could make predictions for such enzymes and compare them to published datasets. Alternatively, the authors could select a challenging model protein (different than an antibody) for optimization by CamSol combination and substantiate the predictions with own experimental data to demonstrate the broad applicability value of their approach.

Reviewer #1 (Remarks to the Author):

This is an interesting manuscript describing a computational pipeline for simultaneously improving folded state stability and solubility of proteins. Importantly, the effectiveness of the pipeline is demonstrated experimentally with different antibody and fragment structures. It is therefore valid for the authors to claim that the method has the potential for impact in biopharmaceutical, and other, applications.

The pipeline largely builds on existing methodology for solubility (CamSol) and stability (Fold-X) predictions, but confirming their applicability when computational and experimental values are compared. There is novelty in the combination of these methods with multiple sequence alignment data (for evolutionary suitability of mutations), and some tweaks of how to put it together.

The pipeline is delivered via a web server.

I agree with the authors that the work is a potentially valuable tool for areas of biopharms and biotech that seek (and likely to be common) improvements in conformational and colloidal stability at the same time.

I feel that the manuscript is rather wordy, but I'm not clear how otherwise it could be constructed to improve the message, perhaps more detail into Suppl Info. Overall it is clear, and the figures good.

We are extremely grateful to the reviewer for the positive assessment of our work, and for the constructive feedback and questions.

Some questions follow.

1. With regard to biopharmaceuticals, the authors mention the driver for increased stability at high concentrations, for storage and administration. Could they discuss the scale of improvement (e.g. mg/ml concentration) that their redesigns could typically achieve?

This is a very relevant question that unfortunately cannot be answered in a simple way. The answer depends very strongly on the protein under scrutiny, and especially on the chosen formulation condition. For example, the mg/mL gain afforded by mutations that affect electrostatic interactions will strongly depend on the ionic strength and pH of the formulation buffer. In our experiments we have selected a suite of widely used developability assays, whose measurements (e.g., midpoint of AMS precipitation or CIC retention time) are well-known to correlate with solubility and hence with the possibility of formulating to higher concentrations. The key advantage of these assays, which is the reasons why they are so widely used, is that they require relatively low amounts of purified protein material, while measuring the improvement in mg/mL in a pharmaceutically relevant formulation would require hundreds of mgs of material (and typically a one-year incubation time). In the revised manuscript we better explain our choice of assay in the discussion section. Because of these challenges, we feel that this type of discussion on the mg/mL improvement is outside the scope of the current work.

2. Early in the manuscript, a class of sites labelled as near surface (and possibly involved in aggregation) are noted. How many such sites make it into the final designs, versus the surface sites?

This answer also depends very strongly on the protein under scrutiny and on the maximum allowed number of mutations. In the revised manuscript (Section "(ii) Selection of candidate mutation sites") we have now added the paragraph:

The typical number of mutation sites in each of these four groups strongly depends on the protein under scrutiny. The final report from the webserver contains a table with all identified candidate mutation sites, which include information on how each site was identified (column “identified from”, see Supplementary Files 1 to 9).

To show that, for each run, this information can be extracted from the final report of the webserver, by looking at the origin of the mutation sites that are contained in the final designs.

3. In the nanobody example, mutation to arginine is seen to be problematic which, as the authors note is a known phenomenon. How would a constraint to avoid this be added to the pipeline / should it?

This is an excellent point; We have decided not to introduce constraints based on liabilities related to stickiness of arginine, chemical degradation hotspots, immunogenicity hotspots etc. as different research groups and pharma companies use their own preferred guidelines depending on the intended application of the therapeutic molecule. In the revised discussion section, we now added the sentence:

At present, users have the option to exclude specific residues from the list of potential substitution targets. So, if any of the final designs contains liabilities, the calculation can be re-run by excluding the relevant residue(s) to get a different set of top-ranking designs (e.g., exclude arginine for specificity, asparagine for deamidation and glycosylation, etc.). Similarly, if any such liability is present in the WT protein, users can add the corresponding site as custom mutation site in the algorithm, so that mutations predicted to improve stability and solubility will be suggested to remove the liability.

4. For Fv example(s), there is a step of checking for interaction of a designed mutation with the antigen binding site, is this part of the pipeline or an add-on. If an add-on, how would it be put into the pipeline?

The antigen-contact analysis was carried out on the WT antibody structures, to identify all sites in contact with the antigen. Then, if any of these sites was mutated in a selected design, we would flag the mutation at that site as potentially disruptive of affinity. We have now made this clearer in the corresponding paragraph of the main text and section of the supplementary material. This analysis is not part of the automated pipeline, but it can be carried out with any standard software for structure visualisation (we have used UCSF Chimera, but Pymol, VMD, MOE and most available molecular visualisation software can readily be used to identify contacts between protein chains). Ultimately, this analysis is exactly what is done routinely to assign the antibody paratope from a bound structure.

5. Do we get sufficient feel for what the mutations are doing, at least in terms of Fold-X $\Delta\Delta G$? e.g. for the nanobody there look to be several charge changes, are these just improving plus/minus network (and then solubility of course too)? Whereas in the Fvs there seem to be more non-charged mutations - what are the anticipated basis of stability change? Is it easy to design proline changes - do these come from Fold-X and MSA together?

The answer is generally yes, but it's very mutation specific. As the reviewer pointed out, the manuscript is already rather wordy, so we feel that discussing the potential molecular basis underpinning the impact of each of the 33 different single amino acid substitutions we validated experimentally is outside the scope of this work. Indeed, some mutations contribute more to CamSol solubility, others more to FoldX energy, some to both. Each mutation does so in different way (for example, proline substitution do come from Fold-X and MSA together, and likely increase the conformational stability by lowering the entropy of the unfolded state).

Users interested to delve deeper into these aspects can do so directly from the webservice output. In the revised manuscript, section “(iii) Single mutational scanning”, we have now added:

The final report from the webservice contains an extract of this table with those substitutions that improve the Mutation Score, and scatter plots showing predicted solubility and stability gains (see Supplementary Files 1 to 9). The full table with all explored substitutions is also provided by the webservice as a .csv file inside the output zip folder. Therein, users can find all details of each attempted mutation, including its calculated contributions to the FoldX total energy (e.g., electrostatics, hydrogen-bonds, solvation, etc.).

Reviewer #2 (Remarks to the Author):

The authors report incremental improvements to an existing software for predicting mutations that improve protein solubility by incorporating a phylogenetic filter that identifies residues that are conserved amongst homologous proteins. By combining this with force field calculations that score stability, the resulting pipeline is expected to enable users to design proteins which have reduced propensity to aggregate. The authors have tested this by applying the algorithm to the sequences of three antibodies in scFvs format (VH-linker-VL) and three single domain nanobodies (VHH). The mutations predicted by the silico analyses have been comprehensively tested experimentally and show that the introduced changes increase thermal stability and solubility in some cases without compromising antigen binding activity. Given these use cases, the work is principally directed at industry users for increasing properties that favour the developability of antibodies as biopharmaceuticals. Although solubility/stability are key parameters, other considerations such as chemical modifications (e.g. deamidation and glycation) are also important in selecting antibody development candidates. The authors indicate that incorporating screening for such sequence-based liabilities forms part of future developments. Although the output of the pipeline provides choices, the experimental scientist using this pipeline still needs to make selections based on other knowledge and experience.

We thank the reviewer for the positive assessment of the work and for recognising the comprehensiveness of our experimental validation.

We agree that our manuscript represents a crucial first step towards the development of an increasingly more far-reaching framework for the optimisation of developability potential, which may include other aspects such as chemical and post-translational liabilities and immunogenicity. We have decided not to introduce constraints based on liabilities related to chemical degradation hotspots, immunogenicity hotspots etc. as different research groups and pharma companies use their own preferred guidelines depending on the intended application of the therapeutic molecule, i.e. some requirements for lyophilized vs. solution formulation, therapeutic vs. diagnostic application and dosage imposed constraints. However, we wish to emphasise that users expert in the field of biologic developability can already use the current implementation to remove such liabilities, albeit this is not fully automated. Sequence-based liabilities present in a WT protein (e.g. deamidation sites, PTM sites) can be entered in the algorithm as custom mutation sites, so that mutations that improve stability and solubility are suggested to remove these liabilities. Similarly, users have the option to exclude specific residues from the list of potential substitution targets. So, if any of the returned final designs contains liabilities, the calculation can be re-run by excluding the relevant residue(s) to get a different set of top-ranking designs (e.g., exclude arginine for specificity, asparagine for deamidation and glycosylation, etc.).

In the revised manuscript (Algorithm section and Discussion section) we have added a few sentences to highlight these opportunities.

Specific comments

1. Antibodies are highly sensitive to concentration and are often required to be formulated at tens of mg/ml. The experimental work reported was carried out at protein concentrations 0.1-1.0 mg/ml. This raises the question of how valid the predictions are for higher antibody concentrations.

This is an excellent point, and we agree with the reviewer that we did not justify very clearly our choice of experimental assays in the previous version of our manuscript. For our experiments we have selected a suite of widely used developability assays, whose

measurements (e.g. midpoint of AMS precipitation, or CIC retention times) are well-known to be predictive of (that is correlated with) solubility and high concentration behaviour. In the discussion section of the revised manuscript, we have now added the sentences:

These are widely used in vitro developability assays, whose measurements are well-known to be predictive of solubility and high concentration behaviour^{43,64,66,85,86}. The key advantage of these assays, which is the reasons why they are so widely used, is that they require relatively low amounts of purified protein material.

We wish to point out that, conversely, measuring the long term-integrity of a pharmaceutically relevant formulation would require hundreds of mgs of material (and typically a ~1-year incubation time). There are many reviews that discuss these and other in vitro developability assays in depth (including a recent one from some of the authors of this work DOI: 10.1007/978-1-0716-1450-1_4). As an example, Jain et al. PNAS 2017 used these assays to characterise most antibodies that were in advanced clinical stages (including approved ones) at the time of their work. We cite these and several other papers on the topic in our manuscript.

2. There are a several other programmes that have been developed to address the developability of antibodies, recently reviewed in Akbar et al 2022 (DOI: 10.1080/19420862.2021.2008790). Although in a different context, the use of a phylogenetic filter in the prediction of mutations in proteins that will improve solubility has also been reported (DOI:10.1016/j.molcel.2016.06.012 DOI 10.1093/bioinformatics/btaa1071). It would be appropriate to comment on the relative merits of different but similar pipelines.

We thank the reviewer for this comment. We have now added a few sentences to the discussion section mentioning the PROSS webserver and three other relevant methods, highlighting some key differences with our pipeline. We believe that an in-depth discussion on the relative merits on the different pipelines and on emerging Machine-Learning approaches would however be more appropriate for a review article, especially as our manuscript is already quite wordy (as pointed out by reviewer #1).

3. The title of the article indicates the applicability of the pipeline to proteins in general but only one example is provided and without any experimental validation. It is suggested that either, further examples are included together with at least an experimental study or that the title is made more specific.

We thank the reviewer for raising this point, which was also raised by reviewer 3. One challenge we have faced when comparing to published work (besides the ProTherm FDR analysis already included in Fig. 1) is that our method automatically identifies specific mutations that should be performed, which are thus unlikely to be those that have been characterised in previous studies. In the revised manuscript, we have circumvented this challenge by benchmarking our predictions with published Deep Mutational Scanning (DMS) data for seven unrelated non-antibody proteins. Our analysis of these data shows that the false discovery rate of our computational procedure is low also for generic proteins, as poorly expressing variants and mutations that disrupt protein function are not shortlisted. See corresponding sections in the revised main text and supplementary method, as well as new figure S10.

4. The choice of nanobodies in the test set is interesting but given that these are typically very soluble and stable proteins, the need to improve this by mutagenesis is unlikely to be a widely adopted application of the software.

We are aware that nanobodies have this reputation, and this is certainly true for many/most nanobodies. However, we find that this is not always the case, based on observations at both

Novo Nordisk and University of Cambridge, and on several discussions with colleagues working elsewhere. Many nanobodies tend to form HMW (high molecular weight) aggregates in HPLC-SEC, and would therefore benefit from solubility improvement. Similarly, a study that analysed a large number (close to 100) of immune-system derived nanobodies found that their thermal stability varied widely, and that most aggregated irreversibly upon heating (Kunz et al. “The structural basis of nanobody unfolding reversibility and thermoresistance.” 2018). It is also relatively well known that it is often a challenge to push the expression yields of some nanobodies to high levels, which can indicate sub-optimal stability and/or solubility. Perhaps, as nanobodies are still emerging as therapeutic molecules, these aspects still need to be systematically investigated. Moreover, nanobodies discovered from synthetic libraries (like Nb.B201 in our work) will be more likely to have sub-optimal stability and solubility, as they haven’t undergone in vivo selection and maturation. Indeed, our algorithm could improve Nb.B201 very substantially (>13°C gain in melting temperature, and >44s gain in CIC RT). Discovery from synthetic libraries offers distinct advantages over immunisation (e.g., it takes less time and money and affords a much easier antigen presentation, including of non-immunogenic antigens), and therefore it’s increasingly employed as a first choice to generate new nanobodies.

5. The run time for the software is admirably fast but log files when runs fail would benefit from some explanation for the non-specialist user.

We have been working on improving this aspect, and since submitting the manuscript have already done several updates to the web server and especially its log. However, it remains difficult to anticipate all possible errors. We anticipate that as users start contacting us reporting errors, we will be able to increasingly improve the log messages.

Reviewer #3 (Remarks to the Author):

The manuscript by Rosace, Bennett et al. presents CamSol combination – a computational pipeline to co-optimize solubility and thermodynamic stability of antibodies (and potentially other proteins) by predicting combinations of favorable mutations. Computational pipelines like this one exist and are of broad interest for rational optimization of antibodies. Two of the essential tools (i.e., CamSol and FoldX) used in the presented approach were published previously and have proven to yield reasonable predictions. To me, the novelty of CamSol combinations comes from the input of phylogenetic information to increase the accuracy of the predictions. The prediction of combinations of mutations is also a step forward. The authors demonstrate with several proteins and dozens of mutants that most of the computational predictions are reasonable. The paper is well written, the method is available via a website which I found user-friendly. The data output from the server is very clear.

We would like to thank the reviewer for these positive comments on our work.

In my opinion, better validation and some additional explanations of the method will be highly beneficial for the manuscript and will increase the value to justify publication in this journal. I listed some comments (not in the order of importance):

- Studies on similar approaches combining colloidal and conformation stability predictions, e.g., TANGO and FoldX (PMID: 2832291), Aggrescan 3D and FoldX (PMID: 31049593), have to be mentioned and the advantages of CamSol combination over these methods should be clearly explained in the discussion. Would the automated Aggrescan 3D/FoldX prediction of mutations overlap to some extent with the individual mutations from CamSol combination? We thank the reviewer for this comment. In the revised manuscript, we added some sentences to the discussion section mentioning the SolubiS webserver and Aggrescan 3D v. 2 (respectively Tango+FoldX and Aggrescan+FoldX), highlighting some key differences with our pipeline. We believe that an in-depth discussion on the relative merits on the different pipelines would however be more appropriate for a review article, especially as our manuscript is already quite wordy (as pointed out by reviewer #1).

As suggested by the reviewer, we ran on the Aggrescan 3D/FoldX webserver the same pdb files (and same excluded sites) used in our work for Nb.B201, Adalimumab and aminoglycoside 3'-phosphotransferase (which is one of the proteins we have used in the new DMS data benchmark, see answer to the last point). Overall, Aggrescan 3D only suggests single mutations, and only to charged residues, without any PSSM-constraint. Hence, albeit there is some overlap in the identified mutation sites, especially those identified on the basis of solubility by our method, there is very little agreement between the two predictions. We include below tables with the mutations suggested by Aggrescan3D, with some comments for each mutation site for the Reviewer's consideration. However, we believe that adding these Aggrescan predictions and a corresponding discussion to the current manuscript would make it too wordy and possibly confusing to readers.

Aggrescan3D + FoldX results

Nb.B201

Mutation	EnergyDiff	AvgScore	AvgScoreDiff	Notes comparing with CamSol Combination (typically one per mutation site)
LR119C	-0.1447	-0.9372	-0.078	in restriction cloning site (LE), no PSSM info Mutation site also identified, PSSM allowed only FRHIS and among these no mutation was shortlisted because FoldX > 0 (as also here EnergyDiff column >0 for mutation to R below)
YE27C	-0.1111	-0.9435	-0.0842	
YK27C	0.0241	-0.9395	-0.0802	
YD27C	0.0379	-0.9417	-0.0824	

YR27C	0.0774	-0.9418	-0.0825	highly conserved, only S would be allowed by PSSM but L is most frequent (not identified as mutation site)	
LR11C	0.1718	-0.9374	-0.0781		
LK119C	0.2029	-0.9335	-0.0742	no PSSM allowed mutations at this site	
LK11C	0.4079	-0.9348	-0.0755		
LE119C	0.4441	-0.9356	-0.0764		
LD119C	0.7485	-0.9353	-0.076		
LE11C	0.8344	-0.9491	-0.0898		
LD11C	1.0738	-0.9505	-0.0912		
GD10C	2.5922	-0.8973	-0.0381		
GK10C	2.8197	-0.8858	-0.0265		
GR10C	2.8695	-0.9012	-0.042		
YK37C	2.8697	-0.8719	-0.0127		
IK28C	2.8713	-0.8658	-0.0065		
YR37C	3.1014	-0.8833	-0.024		Mutation site also identified, but no mutation here is shortlisted because FoldX > 0 (as also here EnergyDiff column >0)
GE10C	3.2528	-0.8983	-0.039		Mutation site not identified as problematic (F only res with higher frequency than Y but leads to worse predicted solubility)
ID28C	3.574	-0.8755	-0.0163		
IE28C	3.7315	-0.8702	-0.0109		
YE37C	3.8561	-0.8511	0.0082		
IR28C	4.1151	-0.8669	-0.0076		
YD37C	5.0842	-0.8452	0.0141		

URL to results: <http://biocomp.chem.uw.edu.pl/A3D2/job/81fbe72e627d7fa/>

Adalimumab

Mutation	EnergyDiff	AvgScore	AvgScoreDiff	Notes comparing with CamSol Combination (typically one per mutation site)
SD103H	-1.1054	-0.7242	-0.0109	Not identified as mutation site (CDR3 position, no solubility issues flagged for serine, but hotspot also present in CamSol profile yet attributed to other residues, in particular to preceding YL both of which are identified as mutation sites)
SE103H	-0.7295	-0.7358	-0.0226	Also identified as candidate mutation site, only mutation to Q allowed by PSSM (not shortlisted as FoldX energy of mutation to Q >0)
SR103H	-0.3986	-0.7365	-0.0233	
SK103H	-0.3847	-0.7349	-0.0216	
VR5H	-0.3479	-0.7459	-0.0326	
VK5H	-0.2511	-0.7379	-0.0247	Also identified as candidate mutation site. Same mutation L102R shortlisted by single-mutational scanning but not beneficial enough to make it in final designs
LR102H	-0.2157	-0.7287	-0.0155	
YR101H	-0.1941	-0.7417	-0.0284	Also identified as candidate mutation site, only HRG allowed by PSSM, nothing shortlisted at this site.
YD101H	-0.1836	-0.7401	-0.0268	in constant domain, no PSSM as design focussed on Fv region
VE5H	-0.0323	-0.74	-0.0267	
LK154L	0.0059	-0.7411	-0.0278	
LE178H	0.0655	-0.7462	-0.0329	
YK101H	0.1573	-0.7397	-0.0264	
LK102H	0.1681	-0.7324	-0.0191	
LK178H	0.2296	-0.7431	-0.0298	
VD5H	0.2466	-0.7372	-0.0239	
LE154L	0.2595	-0.7416	-0.0283	
LR178H	0.2627	-0.7442	-0.031	
LD154L	0.4419	-0.7422	-0.0289	
LR154L	0.4556	-0.7419	-0.0287	
YE101H	0.7454	-0.7402	-0.0269	
LE102H	0.794	-0.7462	-0.0329	
LD178H	0.9969	-0.7526	-0.0393	
LD102H	1.253	-0.7449	-0.0316	

URL to results: <http://biocomp.chem.uw.edu.pl/A3D2/job/7245c922c36ecd3/>

aminoglycoside 3'-phosphotransferase - APH(3')-II

Mutation	EnergyDiff	AvgScore	AvgScoreDiff	Notes comparing with CamSol Combination (typically one per mutation site)
VK242A	-0.3698	-0.8454	-0.0793	Also identified as candidate mutation site. PSSM-allowed are REDKSQNA. Also shortlisted by single mutational scanning.
VK16A	-0.2737	-0.8374	-0.0713	Also identified as candidate mutation site. However, PSSM-allowed are only IL
VR242A	-0.267	-0.8445	-0.0784	Also shortlisted by single mutational scanning.
VR16A	-0.1281	-0.8395	-0.0733	Also identified as candidate mutation site. However, PSSM-allowed are only GSQWF
IE29A	-0.1248	-0.8481	-0.082	
IR29A	-0.1022	-0.8484	-0.0823	
VE242A	-0.0964	-0.8445	-0.0784	Also shortlisted by single mutational scanning.
VE16A	-0.0854	-0.8259	-0.0598	
IK29A	-0.0546	-0.8464	-0.0802	

FD20A	0.0089	-0.8404	-0.0742	Also identified as candidate mutation site. However, PSSM-allowed is only L
FK20A	0.0419	-0.8385	-0.0724	
FE20A	0.0546	-0.8265	-0.0603	
FR20A	0.0946	-0.8408	-0.0747	
ID29A	0.2166	-0.848	-0.0818	
VD16A	0.3223	-0.839	-0.0729	
VD242A	0.4334	-0.8461	-0.08	Not shortlisted (albeit attempted like all mutations at 242) because FoldX energy > 0 (as it is here)
LK81A	0.7333	-0.8149	-0.0487	Also identified as candidate mutation site. However, PSSM-allowed are only IY
VR80A	0.9877	-0.8188	-0.0527	Also identified as candidate mutation site. However, PSSM-allowed are only PI
VK80A	0.9893	-0.818	-0.0519	
LR81A	1.0732	-0.8176	-0.0514	
LE81A	1.3461	-0.8026	-0.0364	
LD81A	2.1683	-0.8174	-0.0513	
VE80A	2.3264	-0.8026	-0.0364	
VD80A	2.649	-0.8155	-0.0493	

URL to results: <http://biocomp.chem.uw.edu.pl/A3D2/job/5b70e59e0ea0cf3/>

- When using different databases (e.g., clinical-stage, OAE, abYsis) for the phylogenetic analysis, some of the predicted mutations overlap, but others do not. Non-expert users of CamSol combination will be faced with the challenge of deciding which databases to use. A better discussion (or maybe a table) highlighting the strong and weak points of using the different databases will be very useful. Comparisons and experimental validations for several antibody mutants optimized via CamSol combination using different sequence databases will be valuable.

We are very grateful to the reviewer for pointing out this potential source of confusion. We have now amended the webserver interface to better guide the MSA selection (see screenshot below) and have added this summary paragraph to the relevant section of the supplementary information:

In summary, the single-domain-VH MSA should be used for nanobodies or other single-domain antibodies. We recommend using the OAS-human MSA for all human antibodies and the OAS-mouse for all mouse antibodies, as these MSAs best recapitulate the diversity of the repertoires they represent. Finally, the post-phase-I MSA can be used in cases where there is a strong need to retain similarity with clinical antibody candidates, as, with 526 sequences, it currently has a more restricted diversity than the OAS-human MSA.

Revised webserver table, with OAS-human now used as default MSA:

In our work, we have run the CamSol combination procedure on Adalimumab (Humira) using both the OAS-Human and the Post-Phase-I MSAs (see Table S3). We experimentally validated the best five-mutation design from each run. These two designs had 3 mutations in common (A23K and W53P on VH and A94P on VL), while the unique mutations were T52S and S55G from the post-phase-I calculation and A40P and S49G from the OAS-Human calculation, all on the VH. Experimentally, we don't find much difference between these two designs, the one obtained from the OAS-human MSA had slightly higher thermal stability (by 0.9 °C), perhaps reflecting the higher diversity, and hence larger 'allowed mutational space', of this MSA.

- FoldX is known to work best with crystal structures as an input. However, the CamSol combination will most likely be used on homology models of antibodies. Can the authors show

an unbiased comparison of the quality of the predictions when using crystal structures versus homology models? Does the server predict the same mutations?

We thank the reviewer for rising this point, as it is likely that some users will use the method starting from modelled structures. We have now performed a new analysis where we compare the results of runs from crystal structures and corresponding models for 19 different antibodies. To make this analysis as unbiased as possible, we selected 19 Fvs that belong to the test set of ImmuneBuilder, and hence were not employed for algorithm training. ImmuneBuilder is the software we have used to obtain the models. We find a very good agreement between predictions carried out starting from the model or from the corresponding structures. To avoid making our manuscript even more wordy, and as this is not the key focus of our work, we cover this new analysis in a small paragraph of the revised discussion section. However, all details of it and a more in-depth discussion of its results can be found in the revised Supplementary Information, and the results are plotted in the new Figure S11.

Overall, approaches to model protein and antibody structures are improving at an unprecedented rate. We expect that, in the near future, there won't be any difference between running our approach on a crystal structure or on a model.

- The melting temperature was used as a proxy for the conformational stability, but measurement of the ΔG of unfolding to validate the predictions would be more accurate. Can the authors provide ΔG values on selected mutants and the corresponding wild-types?

We have now carried out additional experiments of GdnCl chemical denaturation to measure the ΔG of unfolding. We did this for at least two variants per antibody (and 3 for Adalimumab and Nb.b201). As expected, we find that, even if ΔG measurements are affected by larger experimental uncertainties than T_m measurements, there is a perfect agreement in the stability rankings of mutational variants obtained with chemical and heat denaturation. These results are briefly touched on in the revised discussion section, and described in detail in the revised supplementary methods (see new Fig. S12 and Table S4).

- It is not surprising that the ammonium sulphate precipitation did not yield reliable results to validate the predictions of the nanobodies. The cross-interaction chromatography might show some correlations with the predictions, but CIC is not really a measurement for colloidal stability. Can you use a light scattering approach to measure the second osmotic virial coefficient which is the actual thermodynamic parameter that should correlate with colloidal stability? Alternatively, affinity-capture self-interaction spectroscopy is a good proxy for colloidal stability.

We thank the reviewer for these suggestions. The AMS precipitation results we have included have very broad confidence intervals that overlap with each other for many variants. For this reason, we cannot obtain a reliable ranking of relative solubility for all designed variants. However, these results clearly show that the WT nanobody precipitates at a lower AMS concentration than the designed variants, thus supporting the effectiveness of the design. We agree that accurate light-scattering measurements of the second osmotic virial coefficient can be useful. However, such measurements require relatively large amount of purified protein material, as proteins need to be concentrated to concentrations > 1 mg/mL (ideally to 10 mg/mL) and then titrated down. We could not obtain enough material to do these experiments (in addition to those we have done) with our mid-throughput mid-scale expression. We wish to point out that CIC measurements have been shown by many reports to correlate well with measurements of solubility, and in fact CIC is a widely used technique to assess antibody developability potential (see references in the main text).

AC-SINS is a great suggestion, as it would require limited material. However, whereas AC-SINS is well-established for the IgG antibody format relying on a specific anti-Fc antibody being conjugated to the gold nanoparticles, it is not well-validated for other molecular formats such as scFv and nanobodies, where a new capture approach would need to be developed. We are aware that promising data exist on Fab-SINS (Biophysical and Sequence-Based Methods for Identifying Monovalent and Bivalent Antibodies with High Colloidal Stability – PubMed 29154550) but still we do not believe that this method is optimal for the formats in scope here, as it would require extensive assay development and validation, which is outside of the scope of this work.

- In the structures that I optimized with CamSol combinations, there were several interesting predictions. It appears that the colloidal stability of antibodies with basic isoelectric points is often increased by introducing more positive charges (which are known to drive polyspecificity). I did not observe a case where the method chose to optimize an antibody by introducing negative charges to significantly reduce the isoelectric point (although it is known that antibodies with acidic isoelectric points can exhibit high solubility and stability). What is the possible reason for this?

We thank the reviewer for highlighting this potential issue. It is hard to speculate on the possible reasons for this, as they can be many, and they will be specific to the input antibody. Quite generally, positive surfaces would repel each other in solution, hence theoretically improving the colloidal stability (albeit we know too much positive charge can lead to poor specificity). This may be one of the reasons why the method is suggesting adding more charges. Other reasons could be that these are predicted to be more stabilising than alternative substitutions at the same sites, or that they are more frequently observed in the PSSM from the alignment. While polyspecificity is not yet directly predicted by the method, in the revised manuscript (discussion section) we have now added that:

At present, users have the option to exclude specific residues from the list of potential substitution targets. So, if any of the final designs contains liabilities, the calculation can be re-run by excluding the relevant residue(s) to get a different set of top-ranking designs (e.g., exclude arginine for specificity, asparagine for deamidation and glycosylation, etc.). Similarly, if any such liability is present in the WT protein, users can add the corresponding site as custom mutation site in the algorithm, so that mutations predicted to improve stability and solubility will be suggested to remove the liability.

Therefore, in the reviewer's case, it may be useful to re-run the calculations by excluding R and K as potential substitution targets, to see if then the suggested designs would have lower isoelectric point.

- A recent paper presented an *in vitro* approach to select stabilizing mutations in antibodies (PMID: 32286330). I wonder if the computational optimization with CamSol combination will predict some of the mutations reported in this paper.

We thank the reviewer for bringing this work to our attention. We now cite it in the introduction. The authors validate their platform using variants of mAb MEDI1912. They start from the previously reported WFL and STT variants (Fig. 2 therein), and then identify other mutants obtained with error-prone-PCR-driven directed evolution. Error-prone PCR is a powerful unbiased technique, but it can be dangerous to apply it to therapeutic candidates. While it can improve the property under evolutionary selection, it often introduces other liabilities, such as immunogenic motifs or undesired effects on affinity or stability (as the authors observe in their Fig. S7 and Fig. S8 for some of their evolved variants). Conversely, our algorithm applies species-specific PSSM-constraints, which make it unlikely to introduce immunogenicity, and predicts both thermal stability and solubility. Therefore, while a few of

the mutations the authors find with directed evolution may be predicted by our method, there is no reason to expect any statistically significant overlap. Importantly, in our previous work, we have already benchmarked CamSol solubility predictions and aggregation hotspot detection on these variants of MEDI1912, finding a perfect agreement (see Fig. 5 of Sormanni et al. 2017 10.1038/s41598-017-07800-w, where mAb1 and mAb2 are respectively MEDI578 and MEDI1912; panel C uses the same SEC-HPLC data that the authors of PMID 32286330 use in their Fig. 2). As we previously reported a benchmark of CamSol with these mAb variants, we believe that it wouldn't be suitable to report a very closely related analysis in the present work.

- The validations for IgGs were performed by measurements on the scFv. Reformatting an scFv into a Fab can have unexpected effects on protein solubility/stability. Can the authors validate their predictions for several mutants produced as full-length IgGs?

We regret that we are unable to carry out additional rounds of protein production in IgG format and subsequent characterisation. However, we are not aware of a single instance (published or in house) in which the stability or solubility ranking of different mutational variants of the same WT scFv was changed upon reformatting to Fab or full IgG. In other words, the measured property (e.g., the melting temperature) can change, in some cases substantially, when reformatting from scFv to Fab, but the ranking of mutational variants of the same WT scFv is always conserved (e.g., more stable scFv variants yield more stable Fab or IgG variants, provided that the constant domains are always the same). We expect that this will be especially true in our case, as most of our mutations are in the CDR regions and thus far away from the constant domains, and as our Fv regions differ from their WT by a maximum of only 5 mutations. One of the reasons we chose scFv initially is that, in this format, a single unfolding transition is observed facilitating the measurement of stability. Conversely, in IgG format, multiple transitions are typically observed corresponding to the unfolding of the different domains, which can substantially complicate the interpretation of stability measurements.

- It will be highly valuable if the authors show correlations between the CamSol combination predictions and the production yields of the proteins.

We agree with the reviewer that expression yields are a relevant developability readout. However, in our work we have chosen to characterise experimentally an as high as possible number of designed variants. Therefore, we could express these only at relatively small scale and crucially only once. In our experience, expression yields can vary quite substantially from batch to batch under these expression conditions (most of the variability likely comes from the efficiency of the transient transfection), and we do not feel confident in making any claims or drawing any correlation from an N=1 expression experiment. Overall, from this one instance, we found that expression yields under our conditions did not vary much among designed variants of the same WT antibody. Furthermore, expression of early protein batches for research is often done with HEK cells, which is not always representative for development batches that are often expressed in CHO cells using different fermentation systems, and therefore we are not convinced that these data would add sufficient value.

- There is no validation of the method with proteins different than antibodies, although the title implies that the approach also works with other proteins. There is plenty of data on the effect of mutations on enzymatic stability/solubility (except for the used ProTherm database). The authors could make predictions for such enzymes and compare them to published datasets. Alternatively, the authors could select a challenging model protein (different than an antibody) for optimization by CamSol combination and substantiate the predictions with own experimental data to demonstrate the broad applicability value of their approach.

We thank the reviewer for raising this point. One challenge we have faced when comparing to published work (besides the ProTherm FDR analysis already included in Fig. 1) is that our method automatically identifies specific mutations that should be performed, which are thus unlikely to be those that have been characterised in previous studies. In the revised manuscript, we have circumvented this challenge by benchmarking our predictions with published Deep Mutational Scanning (DMS) data for seven unrelated non-antibody proteins. Our analysis of these data shows that the false discovery rate of our computational procedure is low also for generic proteins, as poorly expressing variants and mutations that disrupt protein function are not shortlisted. See corresponding sections in the revised main text and supplementary method, as well as new Figure S10.

Reviewer #1 (Remarks to the Author):

The authors have addressed my comments on the original submission satisfactorily.

Reviewer #2 (Remarks to the Author):

The authors have appropriately addressed reviewer comments in the revised ms.

Reviewer #3 (Remarks to the Author):

The authors adressed my questions adequatly and provided much of the additional data that I asked for. I have no further comments.